# A robust yeast biocontainment system with two-layered regulation switch dependent on unnatural amino acid

Tiantian Chang[1,2,7], Weichao Ding[1,2,3,4,7], Shirui Yan[2,3,4,7], Yun Wang[2,3,4], Haoling Zhang [2,3,4], Yu Zhang [2,4], Zhi Ping [2,3,4], Huiming Zhang[1,2], Yijian Huang[1,2], Jiahui Zhang[1,2], Dan Wang[5,6], Wenwei Zhang[2,4], Xun Xu [2,3,4], Yue Shen [1,2,3,4] ✉ & Xian Fu [2,3,4] ✉

Synthetic auxotrophy in which cell viability depends on the presence of an unnatural amino acid (unAA) provides a powerful strategy to restrict unwanted propagation of genetically modified organisms (GMOs) in open environments and potentially prevent industrial espionage. Here, we describe a generic approach for robust biocontainment of budding yeast dependent on unAA. By understanding escape mechanisms, we specifically optimize our strategies by introducing designed "immunity" to the generation of amber-suppressor tRNAs and developing the transcriptional- and translational-based biocontainment switch. We further develop a fitness-oriented screening method to easily obtain multiplex safeguard strains that exhibit robust growth and undetectable escape frequency ($<\sim10^{-9}$) on solid media for 14 days. Finally, we show that employing our multiplex safeguard system could restrict the proliferation of strains of interest in a real fermentation scenario, highlighting the great potential of our yeast biocontainment strategy to protect the industrial proprietary strains.

With the rapid development of synthetic biology, GMOs with designed functions are increasingly utilized in industrial, biomedical and bioremediation applications[1,2], necessitating the development of biocontainment strategies to prevent propagation of GMO in open environments[3]. In order to limit the unexpected risk of GMOs on natural ecosystems and human health, scientific and industrial communities as well as the government have reached a consensus on the safe utilization of GMOs by following the guidelines first established in the Asilomar conference on recombinant DNA[4]. While physical containment and proper waste management could serve as effective and routine ways to limit the biosafety risk, intrinsic biocontainment is even more important as an additional layer of biological barriers for

controlling the environmental release of GMOs[5]. NIH has presented a guideline stating that the maximum acceptable rate of escape from intrinsic biological containment should below $10^{-8}$ GMO escapees per colony-forming unit (CFU). In addition, biocontainment strategy could be potentially utilized to protect engineered proprietary strains in industrial applications.

Early efforts to develop biocontainment approaches focused on the employment of auxotrophs that require essential compounds for survival[6], toxin/antitoxin-dependent "kill switches" system[7–9], or both[10,11]. However, the auxotroph-based biocontainment strategy could be compromised by the cross-feeding of essential metabolites in the environments. In order to overcome the bottleneck of natural

[1]College of Life Sciences, University of Chinese Academy of Sciences, 100049 Beijing, China. [2]BGI Research, Shenzhen 518083, China. [3]BGI Research, Changzhou 213299, China. [4]Guangdong Provincial Key Laboratory of Genome Read and Write, BGI Research, Shenzhen 518083, China. [5]Guangdong Provincial Key Laboratory of Interdisciplinary Research and Application for Data Science, BNU-HKBU United International College, Zhuhai 519087, China. [6]BNU-HKBU United International College, Zhuhai 519087, China. [7]These authors contributed equally: Tiantian Chang, Weichao Ding, Shirui Yan. ✉e-mail: shenyue@genomics.cn; fuxian1@genomics.cn

auxotrophy, recent works focused on the construction of synthetic auxotrophs whose viability depends on unAA in *Escherichia coli* (*E. coli*)[3,12–19]. This approach is enabled by the orthogonal aminoacyl-tRNA synthetase/tRNA (aaRS/tRNA) pair that site-specifically incorporates an unAA into essential proteins in response to the amber stop codon. Although the intrinsic biocontainment strategy relying on unAA incorporation is useful to develop robust safeguard strains, it remains challenging to design or engineer unAA-dependent essential proteins. For instance, these biocontainment approaches utilized computational design of proteins[14] or elaborate large-scale screening of mutant library[17,18,20]. Therefore, a facile and generic approach to generate unAA-dependent organisms is highly desirable.

The genetically modified budding yeast *Saccharomyces cerevisiae* (*S. cerevisiae*) is widely adopted as eukaryotic chassis for many applications of synthetic biology, particularly for the biosynthesis of commodity chemicals[21]. With the rapid development of DNA synthesis technologies in recent years, whole genome synthesized microorganisms are attracting increasing attentions in various applications due to many useful design features[22,23]. This is probably best exemplified by the synthetic yeast genome project (Sc2.0) strains that could be employed to boost bioproducts or biofuels production by using an inducible evolution system embedded in Sc2.0 yeast[24–31]. Effective biocontainment strategy in *S. cerevisiae* was developed via dual control of essential genes at both transcriptional and recombinational levels[32]. By screening a library of essential genes with different expression levels, the performance of transcriptional-based biocontainment strategy for *S. cerevisiae* was further improved[33]. Another yeast biocontainment strategy was recently developed using engineered fluoride sensitivity[34]. As the final Sc2.0 strain is designed to reassign the TAG stop codon to unAA by systematic swaps of TAG to TAA, it would be ideal to apply the unAA-dependent biocontainment strategy to Sc2.0 strains. Although the final Sc2.0 strain has not yet been completed yet, development of the unAA-dependent biocontainment system would be an important goal for controllable use of this strain. However, whether approach to engineering synthetic auxotrophs is applicable for eukaryotic microorganisms such as *S. cerevisiae* and the potential escape mechanisms remain unknown.

Here, we developed a facile method to construct unAA-dependent eukaryotic microorganism using *S. cerevisiae* as a model. We demonstrated that our biocontainment strategy is applicable to many essential genes with varying expression levels and has minimal impact on the growth rate of resultant yeast strain. By understanding escape mechanisms disclosed in our study, we further optimized our design to reduce the escape frequency of safeguard strains by: (1) increasing the number of TAG codons in selected essential genes, (2) introducing the CRISPR/Cas9-based "immunity" to the amber-suppressor tRNA, and (3) designing the transcriptional- and translational-based multiplex biocontainment switch. In addition, we developed a fitness-oriented library screening method to easily identify optimal permissive residues for unAA substitution, which resulted in multiplex safeguard strains that exhibit robust growth and undetectable escape. Finally, we showed the abundance of safeguard strain is significantly decreased during the second round of two successive 1-liter geraniol fermentation by switching growth conditions (from permissive to non-permissive broth), highlighting the potential use of biocontainment approach to minimize the risk of strain theft by recovering from the residual fermentation broth and prevent contamination in fermentation process caused by the survived cells after incomplete sterilization.

## Results

### Design scheme of engineered yeast dependent on unAA
We aim to establish a generic and facile method to restrict the growth of *S. cerevisiae* in a defined medium containing a specific unAA. This biocontainment approach employs the orthogonal translation system to site-specifically incorporate the membrane-permeable unAA into essential proteins in response to the internal amber stop codon. In this way, the expression of full-length essential protein and the viability of engineered yeast cells depend on the presence of the exogenously supplied unAA in the medium (Fig. 1a). O-methyl-L-tyrosine (OMeY) was used in this study as its orthogonal aminoacyl-tRNA synthetase/tRNA (aaRS/tRNA) pair called LeuOmeRS/tRNA$_{CUA}$ is a commonly used orthogonal pair and shows good efficiency in *S. cerevisiae*[35].

The selection strategy of essential genes is a key factor in the design of the unAA-dependent yeast auxotroph. To avoid inefficient production of the full-length proteins via amber suppression, a process in competition with translation termination, we first tested our biocontainment strategy using essential genes with low expression levels. Based on the ranking of transcript abundance of all essential genes in *S. cerevisiae* according to RPKM (Reads Per Kilobase of transcript per Million reads mapped[36]), two essential genes *CDC4* and *CDC27* with corresponding RPKM values at 31 and 15 were chosen for the initial test (Fig. S1). Another reason to select Cdc4 and Cdc27 proteins is that they perform essential functions by serving as important components of multi-subunit E3 ubiquitin ligases involved in regulation of cell division cycle[37,38] and therefore could not be complemented by the supplementation of metabolic intermediates.

Unlike most of previous studies that focused on the design or engineering of unAA-dependent essential proteins[14–18,20], we aimed to develop a generic strategy for many different proteins. Inspired by a previous work[12] to identify permissive sites for unAA replacement, we simply looked for residues that are highly tolerant of substitution in essential proteins. Ideally, these permissive residues should not play an essential role in proteins. As the levels of conservation correlate with the essentiality of residues for protein activity, the permissive residues might exhibit low conservation among protein homologs. We performed conservation analysis at the residue resolution for the selected Cdc4 and Cdc27 and observed low conservation at the N-terminus of proteins (Fig. S2a). We further analyzed all essential proteins of *S. cerevisiae* and found the conclusion is still valid (Fig. S2b), suggesting that this region is appropriate for OMeY substitution. Except for the N-terminus, we utilized a previously established computational protocol[39], which was originally used to develop unAA-dependent biocontainment system in *E. coli*[12], to find additional 7 permissive residues in the rest region of Cdc4 and Cdc27 proteins for the next step validation.

### Construction and characterization of OMeY-dependent yeast auxotrophs
Based on predicted permissive residues in Cdc4 and Cdc27, we individually replaced corresponding codons with the TAG stop codon via site-directed mutagenesis. We first used a temperature-sensitive (ts) selection system[40] to quickly evaluate our biocontainment strategy. Briefly, the ts mutants were not able to grow at restrictive temperatures unless the introduced LeuOmeRS/tRNA$_{CUA}$ pair could suppress the amber codon to produce full-length proteins encoded by the second copy of essential genes on the episomal vector. We co-transformed two plasmids for simultaneous expression of the LeuOmeRS/tRNA$_{CUA}$ pair and essential gene variants bearing the internal TAG codon into the corresponding ts mutants. Cell viability was then tested in the presence and absence of 1 mM OMeY at both permissive and restrictive temperatures. Except for the ts mutants carrying CDC4$_{TAG270}$ variant (TAG substitution at residue 270, similar nomenclature is used hereafter), all other ts strains covering 10 selected residue sites exhibited the OMeY-dependent growth at the restrictive temperature (Fig. S3). Thus, we demonstrated that our strategy is capable of creating yeast safeguard strains relying on unAA.

To ensure stable single copy of the TAG-containing *CDC4* and *CDC27* genes, we next tried to construct OMeY-dependent yeast auxotrophs using chromosomal integrated gene variants. Based on the spotting assay using ts strains (Fig. S3), four permissive residues were

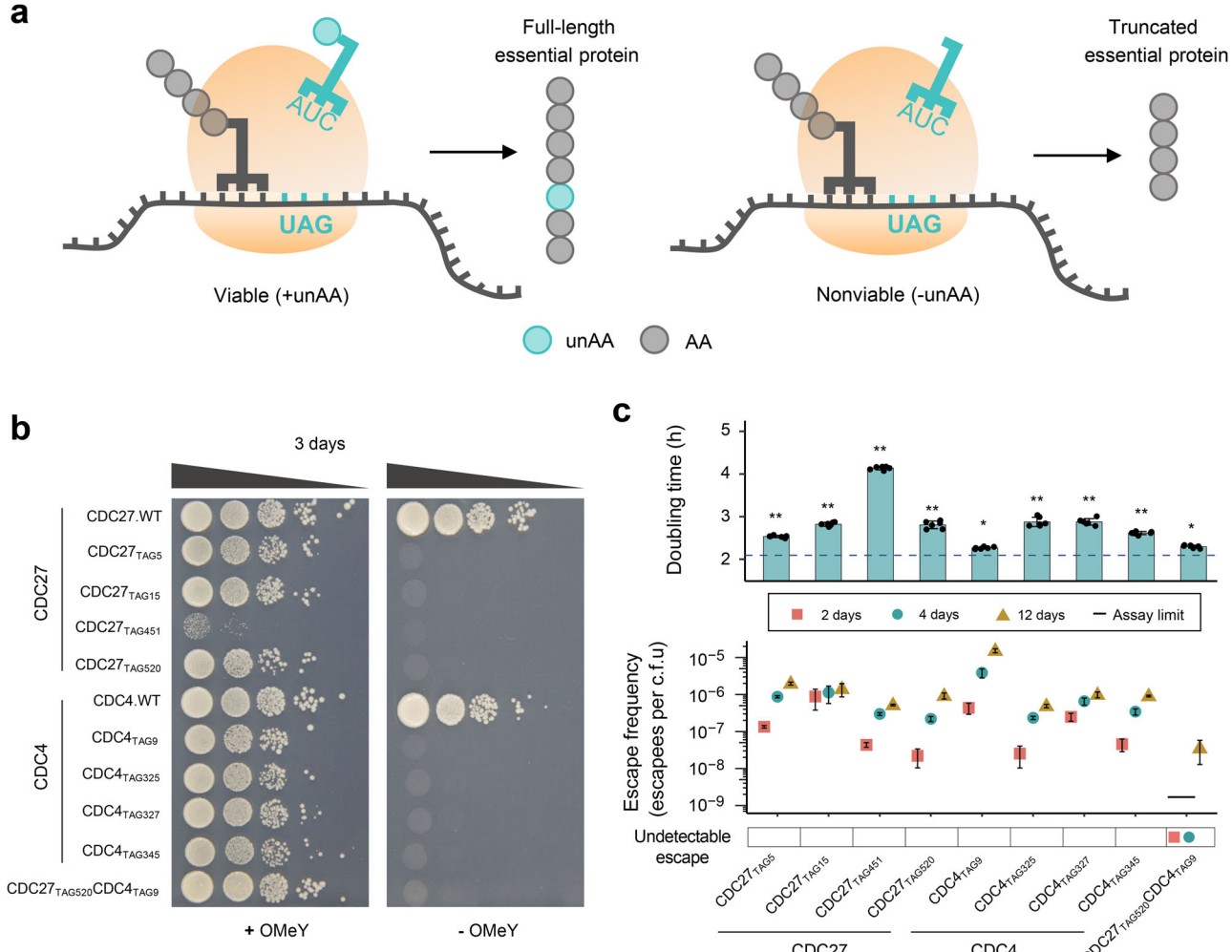

**Fig. 1 | Design and construction of synthetic auxotrophy in which viability of budding yeast depends on the presence of an unnatural amino acid (unAA).**
**a** Strategy used to engineer auxotrophic yeast strain relying on unAA (labeled in blue-green). The expression of full-length essential protein and the cell viability require the presence of OMeY in the media, which is enabled by introducing the orthogonal LeuOmeRS/tRNA$_{CUA}$ pair that site-specifically incorporates OMeY into essential proteins in response to the internal amber stop codon. **b** OMeY-dependent auxotrophs based on Cdc4 and Cdc27 proteins were grown on selective media plates in the presence or absence of 1 mM OMeY. **c** Escape frequencies and doubling times of auxotrophic strains. Doubling times for each strain in SC–Leu medium containing 1 mM OMeY are shown on the top. The dashed horizontal line represents the doubling time of the control strain (BY4741 expressing LeuOmeRS/

tRNA$_{CUA}$ pair). The error bars show the mean and standard deviation (SD) of six biological replicates (*$p < 1 \times 10^{-3}$; **$p < 1 \times 10^{-5}$). $p$ values were determined by two-sided, unpaired Student's $t$ test. The exact $p$ value is provided in Source Data file. Escape frequencies at different times (2, 4, and 12 days, labeled in red square, blue circle, and golden triangle, respectively) are shown on the bottom, which is calculated as colonies observed per CFU plated on non-permissive media plates. For escape assay of each strain, error bars show the mean ± standard error of the mean (SEM) of six samples including three biological replicates that were conducted in duplicate. Assay limit was determined by 1/(total CFU plated) with the assay limit of ~1.7 × 10$^{-9}$ escapees per CFU (see source data for details). Source data are provided as a Source Data file.

selected from Cdc4 and Cdc27 proteins respectively (F9, F325, S327, and L345 of Cdc4; P5, G15, N451, and H520 of Cdc27). To introduce the nonsense mutation via homologous recombination, synthetic DNA fragments of *CDC4* and *CDC27* genes bearing an in-frame TAG stop codon were transformed into the wild-type strain (BY4741) expressing the LeuOmeRS/tRNA$_{CUA}$ plasmid and grown in permissive media containing OMeY. After replica plating on non-permissive media followed by Sanger sequencing verification, we obtained many OMeY auxotrophs. Except for CDC27$_{TAG451}$, all strains grew well on the medium plate containing 1 mM OMeY and showed prohibited growth in the absence of OMeY compared with the wild-type control strain (Fig. 1b). Measurements of their growth rate revealed a certain increase of varying degrees in doubling time compared with the wild-type strain carrying the LeuOmeRS/tRNA$_{CUA}$ plasmid at the same condition (Fig. 1c). To quantify the degree of biocontainment, we measured the escape rate based on the ratio of CFU on non-permissive to permissive

conditions, and all strains exhibited a range of escape frequencies spanning $10^{-5}$ to $10^{-8}$ escapees/CFU during the 12-day observation period (Fig. 1c).

## Evaluating the versatility of OMeY-dependent biocontainment strategy

After demonstrating the feasibility of our strategy to construct OMeY-dependent auxotrophs based on *CDC27* and *CDC4* genes, we next aimed to study whether our approach is applicable to many other essential genes or not, particularly for highly expressed ones. Using the same homologous recombination method described above, we created a series of OMeY auxotrophs using five additional genes (*ALG1*, *ERG8*, *RPN8*, *GDI1* and *NMD3*) with increased RPKM value at 50, 74, 149, 194 and 305, respectively. All obtained strains exhibited OMeY-dependent growth on the solid medium plates (Fig. 2a), with a range of escape frequencies spanning $10^{-5}$ to $10^{-8}$ escapees/CFU (Fig. 2b). To

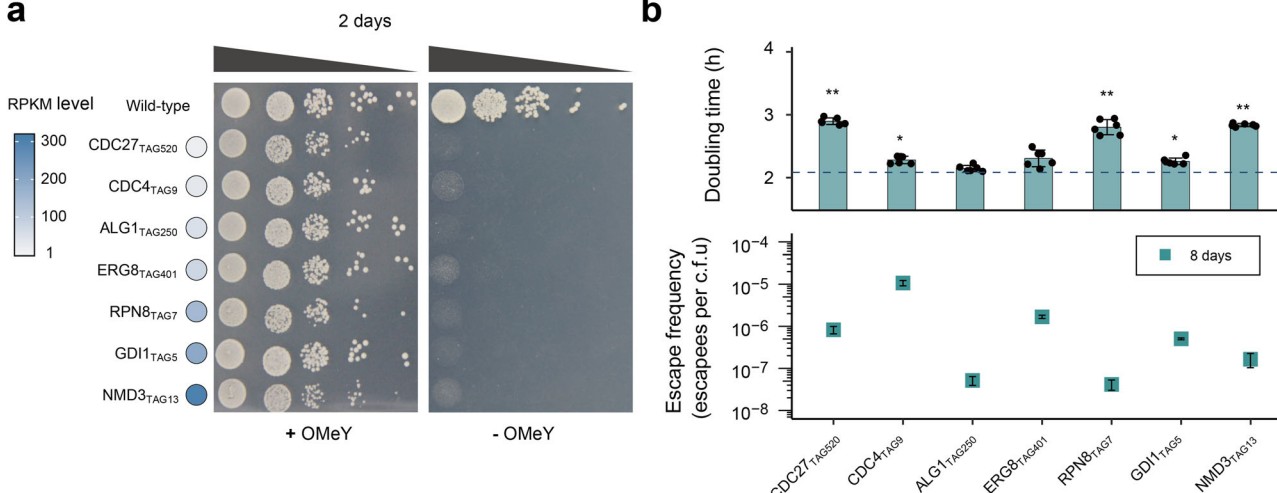

**Fig. 2 | Construction of synthetic auxotrophs based on distinct essential proteins with different RPKM levels. a** OMeY-dependent auxotrophs based on different essential genes were grown on selective media plates in the presence or absence of 1 mM OMeY. The shades of the blue color represent correlates with the transcriptional level of different genes determined by RPKM. **b** Doubling times and escape frequencies of auxotrophic strains. Doubling times for each strain in SC–Leu medium containing 1 mM OMeY are shown on the top. The dashed horizontal line represents the doubling time of the control strain (BY4741 expressing LeuOmeRS/ tRNA$_{CUA}$ pair). For growth rate measurement, the error bars show the mean ± SD of six biological replicates (*$p < 1 \times 10^{-3}$; **$p < 1 \times 10^{-5}$). $p$ values were determined by two-sided, unpaired Student's $t$ test. The exact $p$ value is provided in Source Data file. Escape frequencies of different auxotrophic strains on the day 8 are shown on the bottom, with error bars representing the mean ± SEM of six samples including three biological replicates that were conducted in duplicate. Source data are provided as a Source Data file.

quantitatively evaluate the growth of these synthetic auxotrophs, we measured their doubling time and observed different levels of fitness impairments under the permissive condition (Fig. 2b). For the highly expressed essential gene *NMD3* (RPKM at 305), we tested many permissive sites to obtain OMeY auxotrophs and found they all exhibited obvious growth defect in varying degrees (Fig. S4a). To further understand this, we constructed strains that encode Nmd3–GFP fusion proteins (both wild-type and NMD3$_{TAG13}$) on the chromosome and measured fluorescence signals of cells to quickly evaluate the expression level of Nmd3 protein. We found the fluorescence intensity of strain NMD3–GFP is around 7-fold higher than that of NMD3$_{TAG13}$–GFP (Fig. S4b), which suggested the impaired growth of the *NMD3*-based auxotroph is due to insufficient production of Nmd3 protein by LeuOmeRS/tRNA$_{CUA}$ mediated amber suppression. Taken together, considering that OMeY auxotrophs were successfully constructed based on all seven selected genes with very distinct expression levels (Fig. S1), we showed our biocontainment strategy could be potentially applied to many different essential proteins in *S. cerevisiae*.

## Investigation of escape mutants to understand the underlying mechanisms

We also observed escapees derived from the constructed OMeY-dependent auxotrophs in the escape assay (see "Methods" for details). To further understand the potential mechanisms, we randomly selected eighteen independent escapees (three CDC4$_{TAG9}$, four CDC4$_{TAG325}$, three CDC27$_{TAG5}$, two CDC27$_{TAG451}$, and six CDC27$_{TAG520}$) for whole genome sequencing (WGS) analysis. We found three distinct escape mechanisms were utilized by escapees to grow under the restrictive condition without OMeY (Supplementary Table 1): (1) the mutation of the introduced TAG to a sense codon, as found in one escapee derived from CDC4$_{TAG325}$, (2) the disruption of nonsense-mediated RNA decay (NMD) pathway, a well-known mechanism selectively degrades mRNAs that contain premature stop mutations[41], (3) the occurrence of amber-suppressor tRNAs derived from tRNA$^{Tyr}$, tRNA$^{Leu}$ and tRNA$^{Trp}$.

The appearance of escapees via the first escape mechanism (converting the TAG codon to a sense codon) could be restricted by increasing the number of TAG codons in *E. coli*[12,42]. Consistent with the

previous study, we found that the CDC27$_{TAG520}$CDC4$_{TAG9}$ strain that contains two TAGs in separate essential genes exhibited much lower escape frequency compared with other OMeY-dependent auxotrophs with a single TAG substitution (Fig. 1c). Interestingly, mutations of genes encoding the key regulatory factors (Upf1 and Upf2 proteins) in the NMD pathway were observed in all escapees derived from the CDC4$_{TAG9}$ strain. Since some near-cognate tRNAs and misincorporation of natural amino acids by LeuOmeRS/tRNA in the absence of OMeY might allow stop codon readthrough in yeast[43,44], we hypothesized that the disruption of NMD pathway would lead to the accumulation of mRNAs containing an in-frame UAG stop codon and increased production of full-length Cdc4 protein via nonsense suppression for survival under the restrictive condition. To validate this hypothesis, we separately deleted *UPF1* and *UPF2* genes in the CDC4$_{TAG9}$ safeguard strain and measured cell growth on a medium plate without OMeY. These modified strains showed significantly improved growth compared with the parental CDC4$_{TAG9}$ strain in non-permissive conditions (Fig. S5), confirming that impairment of NMD pathway is indeed an escape mechanism. WGS analysis also revealed the mutations in the anticodon loop of tRNA$^{Tyr}$, tRNA$^{Leu}$, and tRNA$^{Trp}$ to form amber suppressor tRNAs that could decode the UAG stop codon by following Watson-Crick base pair rules. The amber suppressor tRNA derived from tRNA$^{Tyr}$ was revealed to be the most commonly observed one. Interestingly, we also found previously undiscovered tRNA$^{Leu}$ and tRNA$^{Trp}$ gene loci that could be mutated to form amber suppressor tRNAs in *S. cerevisiae* (Supplementary Table 1).

## Reducing escape frequency by exploring CRISPR-Cas9 system to restrict the occurrence of amber suppressors derived from tRNA$^{Tyr}$

Next, we aimed to design experiments to reduce escape frequency based on our understanding of mutagenic routes that undermine the biocontainment strategy in yeast. The G34 > C substitution in the anticodon loop of tRNA$^{Tyr}$ is the most common mutation observed in all escape mutants. Thus, we focused on restricting the generation of tRNA$^{Tyr}$-derived amber-suppressor by employing the well-established CRISPR/Cas9 system. Specifically, we utilized a Cas9 variant (SpCas9-

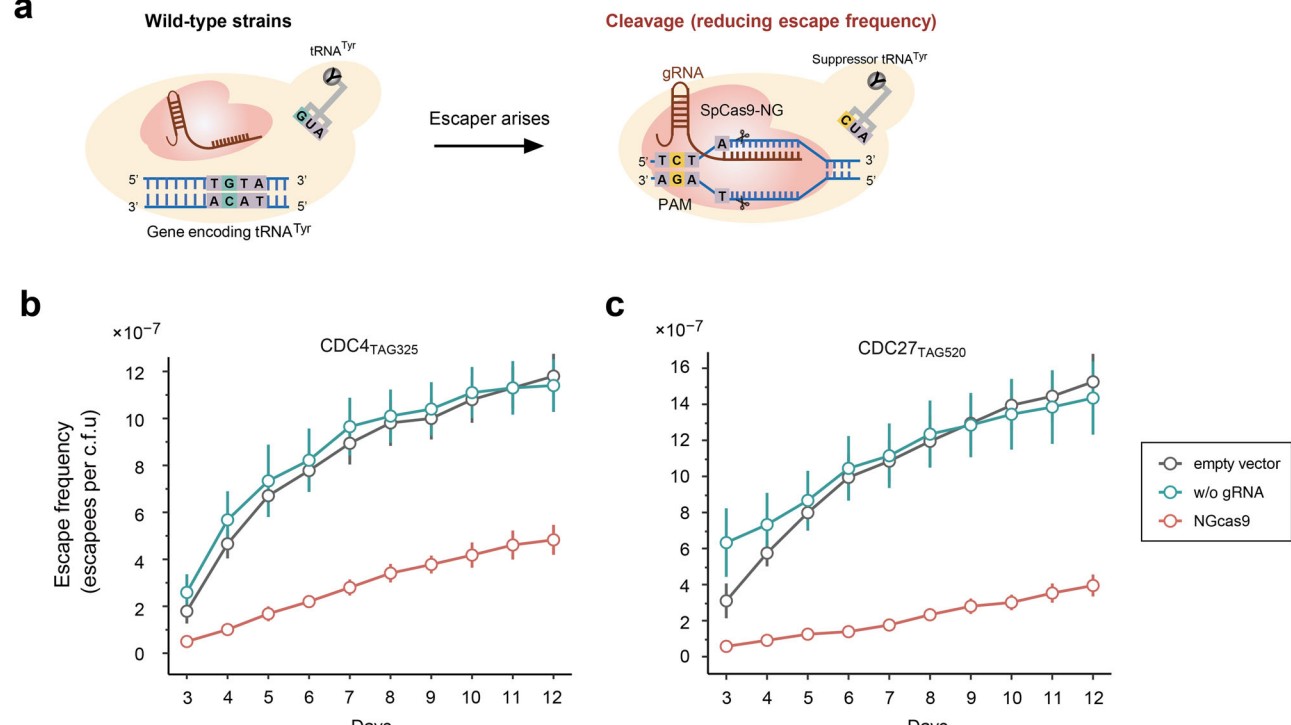

**Fig. 3 | Design and test of the strategy for reducing escape frequency by exploring CRISPR/Cas9 system to restrict occurrence of amber suppressor derived from tRNA^Tyr. a** Schematic diagram showing the design of CRISPR/Cas9-based "immunity" to the amber-suppressor tRNA derived from tRNA^Tyr. The escapee strain carries the C > G mutation on the non-template strand of the gene encoding tRNA^Tyr, which results in the production of tRNA^Tyr(CUA) with the G34 > C mutation in its anticodon. The 5'-AGAT-3' sequence (G is labeled in yellow) in the genome of the escapee serves as the PAM site for SpCas9-NG when programmed with the designed gRNAs. In contrast, the wild-type strain has the corresponding 5'-ACAT-3' sequence (C is labeled in blue-green) and lacks the PAM site. Thus, if an

escapee arises within the population, it would be exposed to double-strand breaks enabled by the SpCas9-NG (labeled in light red) while the wild-type strain would be resistant to this event. Effect of CRISPR/Cas9-based "immunity" on escape frequency of OMeY-dependent auxotrophs based on CDC4$_{TAG325}$ (**b**) and CDC27$_{TAG520}$ (**c**) during the 12-day observation period. The expression of SpCas9-NG was induced by 2% (wt/vol) galactose. Strains that express the empty vector and the SpCas9-NG plasmid without the gRNAs serve as the controls. Escape frequencies were monitored from 3 to 12 days with error bars showing the mean ± SEM of six samples including three biological replicates that were conducted in duplicate. Source data are provided as a Source Data file.

NG) that can recognize the relaxed 5'-NGN-3' protospacer adjacent motif (PAM)[45,46]. Two types of tRNA^Tyr genes, with a total copy number of eight, are encoded in the nuclear genome of *S. cerevisiae*, whose sequences only differ in their introns (Fig. S6a). We designed two guide RNAs (gRNAs) that specifically target the predicted sequences of amber suppressor tRNA^Tyr(CUA) adjacent to the PAM site that contains the G34 > C mutation in tRNA^Tyr (Figs. 3a and S6a, b). As only the mutated gene encoding tRNA^Tyr(CUA) has the 5'-NGN-3' PAM site, we hypothesized that SpCas9-NG programmed with the designed gRNAs would lower the escape frequency by eliminating escapees that carry the G34 > C mutation, which we refer to as immunity to tRNA^Tyr(CUA) occurrence. Expression of SpCas9-NG was driven by the galactose-inducible *GAL1* promoter to allow regulation of this system (Fig. S6c).

We tested our design using two biocontainment strains (CDC4$_{TAG325}$ and CDC27$_{TAG520}$) that were found to generate escapees via the occurrence of tRNA^Tyr(CUA). These strains were transformed with the CRISPR/Cas9 or control plasmid and their escape rates were monitored during the 12-day observation period. As expected, we found the activation of our designed CRISPR/Cas9 system can reduce the escape rate by several folds comparing to the control groups without editing activity by this CRISPR-Cas9 system (Fig. 3b, c). This immunity effect is more evident for CDC27$_{TAG520}$ (4.0 versus -15 × 10^{-7} escapees/CFU for groups with and without the use of the CRISPR-Cas9 system at day 12) compared to CDC4$_{TAG325}$ (around 4.8 versus -11 × 10^{-7} escapees/CFU for groups with and without the use of the CRISPR-Cas9 system at day 12), which could be explained by our finding that the

frequency of tRNA^Tyr(CUA) occurrence in CDC27$_{TAG520}$ is higher than that of CDC4$_{TAG325}$ (Supplementary Table 1). To confirm the editing activity of our introduced immunity to tRNA^Tyr(CUA), we performed WGS analysis for some of escapees expressing the CRISPR/Cas9 plasmid, and detected leftover scar interrupts at the specific tRNA^Tyr location directed by the designed gRNAs (Fig. S6d and Supplementary Table 2).

## Reducing escape frequencies by combining transcriptional and translational control of essential genes

As entirely different escape mechanisms are utilized to overcome transcriptional-based and unAA-dependent biocontainment strategies, we hypothesize that the multiplex biocontainment switch would significantly increase the difficulty of yeast cells to escape and thus would further reduce the escape frequency (Fig. 4a).

We first tested the transcriptional control of three essential genes (*RPC11*, *SKP1*, and *RPS3*) by replacing their native promoters with the inducible *GAL1* promoter and measured cell growth on the agar plate with and without 2% (wt/vol) galactose. The *RPC11* safeguard strain was successfully constructed in a previous study[32]. Here we tested two additional genes, namely *SKP1* and *RPS3* (RPKM values at 371 and 1329 respectively), that exhibit much higher transcription levels than *RPC11* (RPKM value at 34); and their transcriptional levels are in top 10% of all essential genes (Fig. S1). Selection of these two highly expressed genes was because we hypothesized that leaky expression of highly expressed genes should not be enough to support the cell growth under the non-permissive condition. All three strains grew as expected on the

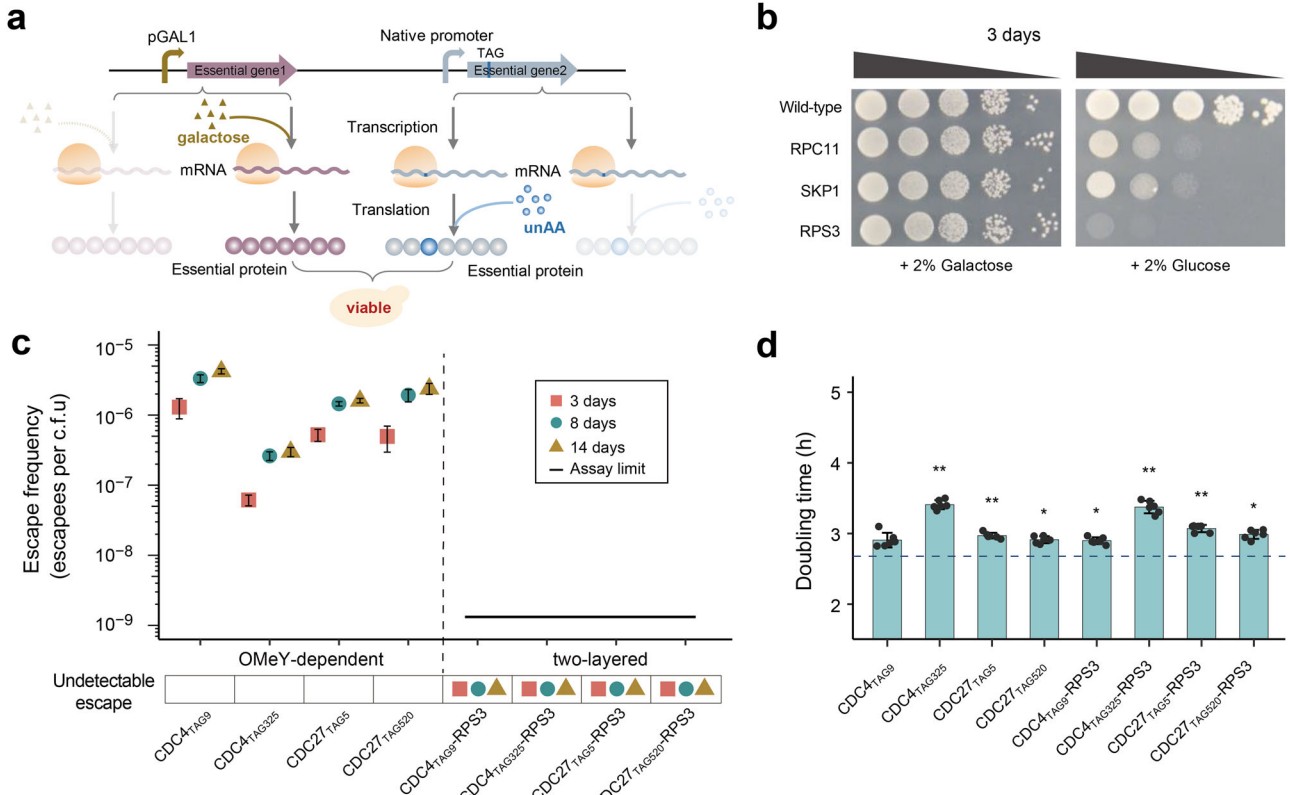

**Fig. 4 | Design and construction of two-layered safeguard strains. a** Schematic diagram showing the strategy of transcriptional- and translational-based multiplex biocontainment switch. Two essential genes are regulated separately. The transcriptional-based control is achieved by replacing the native promoter of one essential gene with the galactose-inducible promoter *pGAL1*. The translational-based control is achieved by unAA-dependent biocontainment strategy. Both galactose (labeled as khaki triangle) and unAA (labeled as blue circle) are required for the growth of two-layered safeguard strains. The TAG substitution, promoters and essential genes are indicated. **b** Safeguard strains based on the transcriptional regulation of three essential genes (*RPC11*, *SKP1*, and *RPS3*) were grown on selective medium in the presence or absence of 2% (wt/vol) galactose. The wild-type BY4741 expressing LeuOmeRS/tRNA$_{CUA}$ pair serves as the control. **c** Escape frequencies of OMeY-dependent auxotrophs and two-layered safeguard strains at day 3, 8 and 14

are indicated as red square, blue circle and yellow triangle, respectively. Escape frequencies were calculated as colonies observed per CFU plated non-permissive media plates (without OMeY and with 2% glucose). Assay limit was determined by 1/(total CFU plated) with the assay limit of $-1.1-1.3 \times 10^{-9}$ escapees per CFU (see source data for details) and the black line represents undetectable growth. Error bars show the mean ± SEM of six samples including three biological replicates that were conducted in duplicate. **d** Doubling times of OMeY-dependent auxotrophs and two-layered safeguard strains in SC–Leu medium containing 1 mM OMeY and 2% (wt/vol) galactose, with dashed horizontal lines showing the doubling times of the control strain (BY4741 expressing LeuOmeRS/tRNA$_{CUA}$ pair). The error bars represent the mean ± SD of six biological replicates (*$p < 1 \times 10^{-3}$; **$p < 1 \times 10^{-5}$). $p$ values were determined by two-sided, unpaired Student's $t$ test. The exact $p$ value is provided in Source Data file. Source data are provided as a Source Data file.

SC–Leu plate supplemented with 2% (wt/vol) galactose and stopped growing on the non-permissive solid medium containing glucose (Fig. 4b), with the escape frequencies of $10^{-4}$ to $10^{-7}$ (Fig. S7). As expected, we found that strains with higher RPKM values have lower escape frequencies. Based on the cell fitness measured by doubling time and the escape frequency of all three transcriptional-based safeguard strains (Fig. S7), we chose *RPS3* for constructing the multiplex biocontainment system.

We next replaced the native promoter of *RPS3* with the inducible *GAL1* promoter in four OMeY auxotrophs (CDC4$_{TAG9}$, CDC4$_{TAG325}$, CDC27$_{TAG5}$, and CDC27$_{TAG520}$). As anticipated, we found that combining transcriptional and translational control of separate essential genes could significantly reduce escape frequency below the detection limit ($<10^{-9}$) (Fig. 4c). Specifically, no escapee was observed after $\sim9 \times 10^{8}$ viable cells were plated across four non-permissive plates. These two-layered safeguard strains exhibited comparable growth to the corresponding parental strains (OMeY-dependent auxotrophs) under the same permissive condition (1 mM OMeY and 2% galactose) (Fig. 4d). In addition, the two-layered safeguard strain derived from CDC4$_{TAG9}$ showed a minor fitness impairment compared to the wild-type strain (Fig. 4d). These findings demonstrated that combining transcriptional and translational control of separate essential genes is

an effective approach to significantly reduce escape frequencies and has a small impact on yeast growth.

## Fitness-oriented screening for permissive residues

Ensuring robust growth of safeguard strains is an important consideration for developing intrinsic biocontainment strategy. We observed OMeY incorporation at different permissive site in the same protein affects the growth rate of the OMeY-dependent strains (Fig. 1c and CDC4$_{TAG9}$ versus CDC4$_{TAG325}$ in Fig. 4d). Thus, we next aimed to develop a facile method to rapidly identify the optimal permissive residues for unAA replacement that has minimal impact on the growth of multiplex safeguard strain. As the N-terminal sequence of proteins exhibits high flexibility (Fig. S2a), we constructed a variant library containing an internal TAG codon for OMeY substitution at individual N-terminal residue of Cdc4 and Cdc27 (from position 4 to 20). The DNA library was then transformed and integrated into the genome of the transcriptional safeguard strain (P$_{GAL1}$-RPS3) for fitness screening in both the permissive and non-permissive media and growth measurement. Finally, desired multiplex safeguard strains showing robust growth would be identified according to the doubling time determined by growth curve measurement (Fig. 5a).

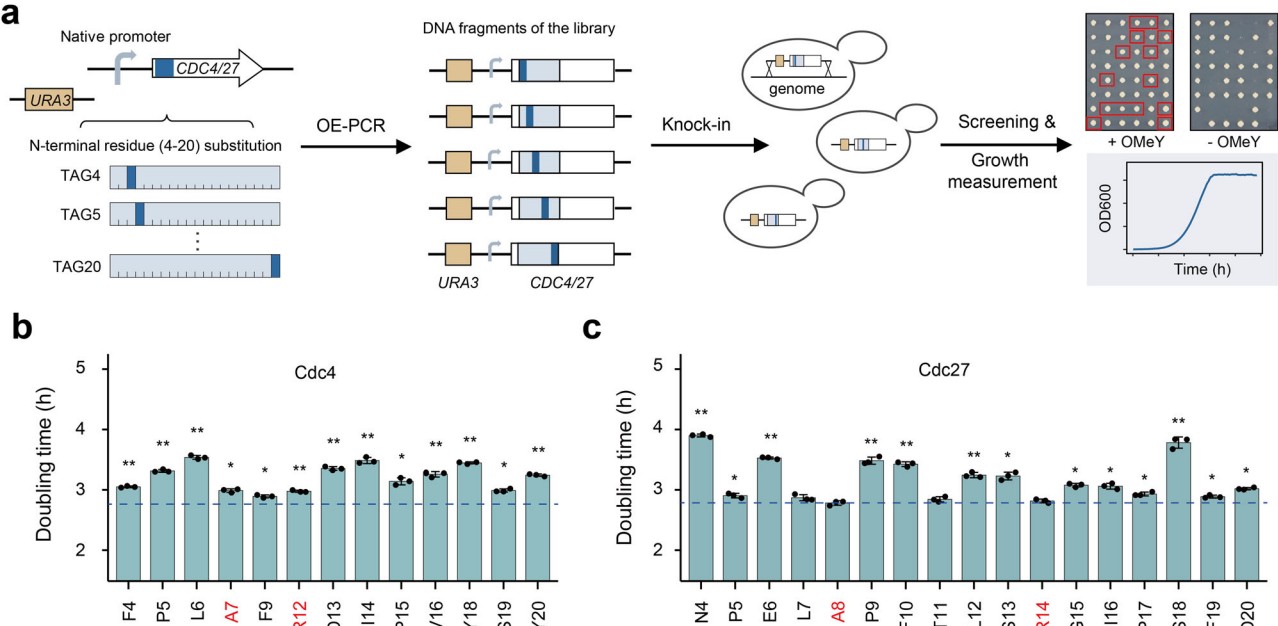

**Fig. 5 | Fitness-oriented screening method to identify permissive residues for OMeY substitution at the N-terminus of Cdc4 and Cdc27 proteins. a** Schematic diagram showing the workflow of the fitness-oriented screening method. To replace the codon corresponding to individual residues at the N-terminus (from residue 4 to 20) of *CDC4* and *CDC27* genes with TAG, synthesized DNA libraries were transformed and integrated onto the genome of the transcriptional safeguard strain ($P_{GAL1}$-RPS3) via *URA3* marker-assisted homologous recombination. Then, desired two-layered safeguard strains showing robust growth could be identified based on the growth curve analysis from all obtained OMeY auxotrophs. Doubling time of two-layered safeguard strains derived from Cdc4- (**b**) and Cdc27-based (**c**) OMeY auxotrophs respectively. The blue dotted line represents the doubling time of the control strains ($P_{GAL1}$-RPS3 strain containing the *URA3* marker integrated at the upstream of wild-type *CDC4* and *CDC27* genes respectively). The error bars show the mean and SD of three biological replicates (*$p < 1 \times 10^{-2}$; **$p < 1 \times 10^{-4}$). $p$ values were determined by two-sided, unpaired Student's $t$ test. The exact $p$ value is provided in Source Data file. The $x$-axis shows different residues that are tolerant for OMeY substitution; four residues were labeled in red to represent corresponding safeguard strains that were confirmed to have undetectable escape when monitored for 14 days (assay limit is -1.1 and -1.2 × $10^{-9}$ escapees per CFU for (**b**) and (**c**), respectively). Source data are provided as a Source Data file.

We randomly collected 124 and 146 Cdc4- and Cdc27-based OMeY auxotrophs respectively. The PCR-based target sequencing of selected safeguard strains revealed the majority of target codons could be replaced by the amber codon (Fig. S8), demonstrating the high tolerance of N-terminal residues for substitution. As expected, representative safeguard strains based on distinct permissive residues exhibited different doubling time (Fig. 5b, c). To test whether the growth differences would scale to fermentation conditions, we measured the growth of representative multiplex safeguard strains cultured in 80 ml condition. We found the fold changes of doubling time relative to the wild-type growth are highly comparable between different growth conditions (0.2 ml versus 80 ml) as shown in Fig. S9. We obtained several two-layered safeguard strains showing robust growth compared with the corresponding control strains ($P_{GAL1}$-RPS3 strain containing the *URA3* marker integrated at the upstream of WT *CDC4* and *CDC27* genes respectively). For example, replacing the residues 7, 9, and 12 of Cdc4 and the residues 7, 8, 11 and 14 of Cdc27 by OMeY resulted in safeguard strains exhibiting near-control fitness (Fig. 5b, c). We also found that supplementation of 0.2 mM OMeY was the minimal requirement for optimal proliferation based on the growth curve at different concentrations of OMeY (Fig. S10). In addition, four representative safeguard strains with high fitness were confirmed to have undetectable escape activities after a 14-days monitoring (Fig. 5b, c). Taken together, we developed a facile and generally applicable method to screen safeguard strains exhibiting robust growth and undetectable escape.

The high fidelity of the orthogonal translation systems is pivotal for synthetic auxotrophy applications. In order to ensure the orthogonality of LeuOmeRS/tRNA$_{CUA}$ pair, we further measured the escape frequencies of two representative OMeY auxotrophs that have low detectable or undetectable escape rates in the presence of 5 mM

leucine or 2.6 mM tyrosine (the maximal solubility in water). The selection of these two natural amino acids is because tyrosine is structurally similar to OMeY and leucine might be recognized by LeuOmeRS that is modified from *E. coli* leucyl-tRNA synthetase. We found CDC27$_{TAG5}$ exhibited close escape frequency under 2.6 mM tyrosine and 5 mM leucine conditions (5.1 and 5.5 × $10^{-7}$ escapees/CFU at day 4, respectively, Fig. S11) compared with the normal condition (8.7 × $10^{-7}$ escapees/CFU at day 4, Fig. 1c), and no escapee was observed after plating ~0.8 × $10^{9}$ CFUs of multiplex safeguard strain CDC27$_{TAG5}$-RPS3 on non-permissive plates supplemented with abundant tyrosine and leucine (Fig. S11). Thus, these data suggested our biocontainment system would still be robust in a real fermentation scenario or the environments, where leucine or tyrosine may be present in higher abundance.

## Evaluating the biocontainment performance of our synthetic safeguard strains

Intrinsic biocontainment is considered an effective way to protect proprietary strains that are the key intellectual property of biomanufacturing industries[47]. Obtaining the proprietary strains from the environment would be almost impossible due to the fact that unAAs do not exist in the environment. The fermentation broth containing the proprietary strains would be the main resource for intentional strain theft by recovering from a very small amount of it. We hypothesized that the controllable growth of safeguard strains would be possible to restrict the chance of stealing strains from broth. Besides, intrinsic biocontainment might be useful to prevent contamination in the fermentation process due to residual cells survived from incomplete sterilization. In order to demonstrate the application of our multiplex biocontainment strategy to protect proprietary strains and prevent contamination, we monitored the dynamic change of the

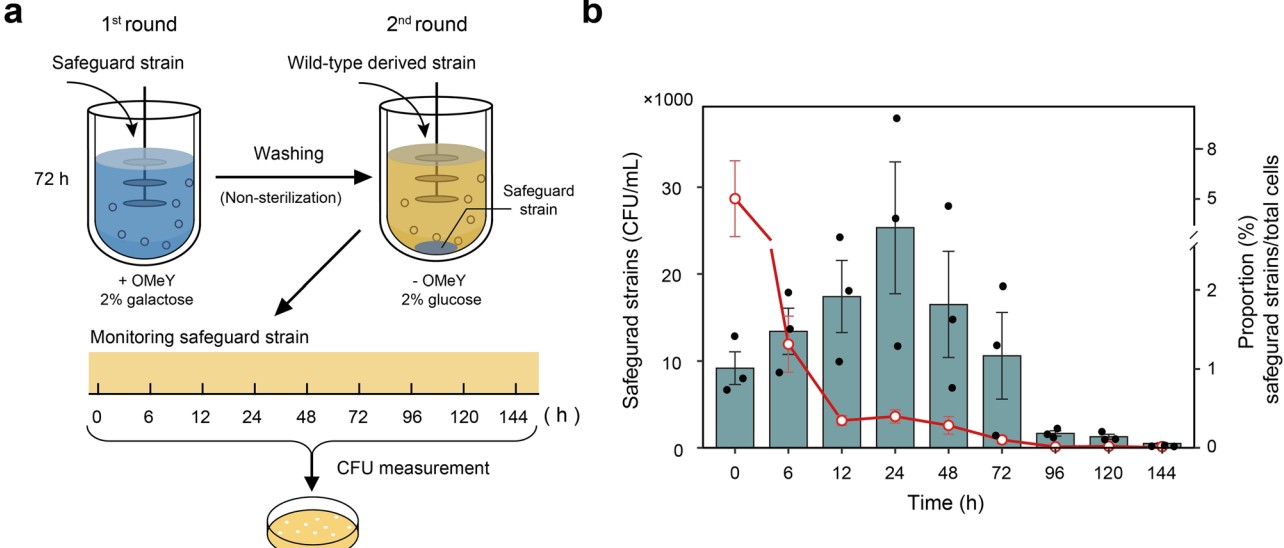

**Fig. 6 | Dynamic change of the residual safeguard strain during the second round of two successive 1-liter geraniol fermentation after switching growth conditions. a** Schematic diagram showing the experimental design. The multiplex safeguard strain (yXF281) was used in the first round of fermentation (72 h, indicated in blue) supplemented with 1 mM OMeY and 2% (w/v) galactose, followed by the second round of fermentation (144 h, indicated in khaki) using the wild-type derivative strain (yXF282) grown in non-permissive broth (without OMeY and 2% glucose) for yXF281. Both yXF281 and yXF282 strains were modified to produce geraniol. The medium used for different rounds of fermentation are indicated. Before the second round of fermentation, the equipment was washed with 3-liter sterilized water (the total volume of fermenter). The fermentation broth from the second round of fermentation was sampled at different timepoints to determine the CFU of yXF281 and total cells (yXF281 and yXF282). Viable titer of yXF281 was determined by using selective medium plates supplement with 1 mM OMeY, 2% (w/v) galactose, and G418 (200 μg/ml). See "Methods" for details. **b** Measurement of the CFU of yXF281 and its proportion in the population at different timepoints. The cyan bar represents the CFU/ml of yXF281 and dot connected by red line represents its proportion. The fermentation experiments were conducted in triplicate and the error bars represent the mean ± SEM of three biological replicates. Source data are provided as a Source Data file.

residual safeguard strain during the second round of two successive 1-liter geraniol fermentation after switching growth conditions. Specifically, the modified multiplex safeguard strain with the ability to produce geraniol (yXF281) was used in the first round of fermentation for 72 h. After draining the fermentation broth and washing cells with 3-liter sterilized water, the second round of fermentation was carried out by the wild-type strain derivative (yXF282) grown in the non-permissive medium for the safeguard strain (Fig. 6a). The initial CFU of residual safeguard strain per ml of fermentation broth was ~9000, which accounted for 5.0% of the whole cell population. We noticed that the number of viable safeguard strain underwent a small rise during the first 24 h, probably due to the residual growth, and then dropped to 204 CFU/ml at the end of fermentation for 144 h (Fig. 6b). At this time point, the proportion of safeguard strain was diluted to an extremely low level at 0.004%, which could be explained by the prohibited proliferation of the safeguard strain under the non-permissive condition and the exponential growth of the wild-type strain.

In our assay, different essential compounds including OMeY, galactose and an antibiotic G418 were all needed for the selective recovery of the multiplex safeguard strain. In fact, the total number of all combinations of required compounds are extremely huge as ~60 unAA have been genetically encoded in yeast cells[48,49], ~40 different compounds have been developed for transcriptional regulation of yeast genes[50,51], and ~25 selection conditions corresponding to distinct resistance markers are available in budding yeast[52]. We further calculated the probability of successfully sub-culturing the safeguard strain from 1 ml broth, a common sample volume used for routine analysis[53], by testing all 60,000 (=60 × 40 × 25) combinations. Our data showed this probability is only 3.9% after 144 h of fermentation, suggesting that the chance of strain theft would be at a very low probability. Overall, our results highlight the effectiveness of our biocontainment strategies to prevent strain recovery and potential espionage activities in a real fermentation cycle.

## Discussion

The creation of synthetic safeguard strains whose survival is dependent on unAA is an effective method to introduce orthogonal barriers between GMOs and the environment and to reduce the risk of spreading them into the open environment[3]. Despite considerable progress in bacteria, construction of unAA-dependent eukaryotic microorganisms such as *S. cerevisiae* has not been reported yet. In this study, we developed a generic approach to engineering a robust yeast biocontainment system for the first time. Our strategy employed the orthogonal aaRS/tRNA pair (LeuOmeRS/tRNA_{CUA}) to site-specifically incorporate OMeY into essential proteins in response to the UAG stop codon. Importantly, we showed our biocontainment strategy has great potential to be applied to most of essential proteins in *S. cerevisiae* including those highly expressed ones. We also demonstrated engineering of unAA-dependent biocontainment has minimal impact on the fitness of *S. cerevisiae* strains by examining the growth rate.

The selection of permissive sites for unAA substitution is a key factor that affects the fitness and the escape frequency of unAA-dependent safeguard strains. Here, we developed a facile method to rapidly identify the optimal permissive residues for unAA substitution by systematically replacing individual amino acid residues at the N-terminus of essential proteins, a region exhibiting low conservation (Fig. S2), followed by the growth measurement of the safeguard strain library. Compared with previous studies that require advanced computational design[14] or extensive protein engineering via directed-evolution methods[17,18,20], our fitness-oriented screening method to create unAA-dependent proteins provides an alternative solution and might have lower technical barrier. By using a DNA library for OMeY substitution at the N-terminus of selected proteins, we could rapidly obtain synthetic auxotrophs with robust growth within 1 week. As the library-based screening method to determine permissive residues requires minimal prior knowledge of the target protein, our strategy would serve

as a generic approach for developing biocontainment system using many different essential genes.

The escape mechanisms that disable the unAA-dependent biocontainment system in *S. cerevisiae* were previously unexplored. In this study, conversion of TAG codon to a sense codon and occurrence of amber-suppressor tRNAs were disclosed when developing the synthetic auxotrophs using *S. cerevisiae*. Consistent with the previous study[12,42], we found that the first escape route could be restricted by increasing the number of TAG codons introduced into separate essential genes. In addition, we harnessed the CRISPR-Cas9 system to specifically minimize the generation of amber suppressors derived from tRNA[Tyr], highlighting the feasibility to construct safeguard strains with designed "immunity" to amber suppressor occurrence. However, it should be noted that this approach had relatively limited impact on the biocontainment system, possibly due to its specific targeting on tRNA[Tyr], but not other escape mechanisms. As different kinds of suppressor tRNAs (such as tRNA[Leu] and tRNA[Trp] derivatives) could occur in escapees, it might be possible to design an array of multiple "immunity" devices (e.g., gRNAs targeting many different suppressor tRNAs) to further optimize this strategy. A unique finding of this study is that disruption of NMD pathway, a surveillance system exists in all eukaryotes[54], represents as one of the major escape routes when developing OMeY-dependent *S. cerevisiae* strains. This escape mechanism could be explained by increased stop codon readthrough due to accumulation of UAG-containing mRNAs in NMD mutant strains. Supporting this idea, a previous study found the incorporation efficiency of unAA in response to the UAG codon is significantly enhanced in the NMD-deficient yeast strain[55]. As it is challenging to prevent the mutation of NMD pathway, we developed a two-layered biocontainment system by combining transcriptional and translational control of essential genes, which could significantly reduce the escape frequency ($< -10^{-9}$). Despite our success in the lab, the limitations of transcriptional kill switch are galactose is naturally available that would break the biocontainment system and glucose is usually the desired carbon source in a real fermentation, which would lead to cell death. In addition to the transcriptional-based switch, we hypothesize that various multiplex safeguard strains could be developed by the joint use of unAA-dependent biocontainment strategy and other well-established methods, such as kill switches based on toxin/antitoxin[56] and synthetic gene circuits[57]. Overall, our study provides useful guidance to engineer and optimize the unAA-dependent biocontainment of *S. cerevisiae* and other eukaryotic cells.

Preventing the industrial espionage via intentional recovery of engineered yeast strains used for large-scale fermentation becomes a growing consensus but is technically challenging. Development of unAA-dependent safeguard strains provides a promising solution to minimize the risk of espionage activities in industrial applications. Here, we tested this idea in a real scenario and quantitatively evaluate the remnant of the safeguard strains during a cycle of geraniol fermentation after switching growth conditions. We found the number of multiplex safeguard strain dropped to 204 CFU/ml under the non-permissive condition after 144 h fermentation, and therefore would significantly reduce the risk of intentional strain recovery. In addition, as only the strain developer knows the permissive condition for synthetic auxotrophs, the protection of media formula would also serve as an important supplement to the protection of the proprietary strains. It is worth pointing out that biotechnology companies could develop the customized biocontainment of proprietary strains by engineering their in-house orthogonal aaRS/tRNA pairs that recognize unique unAA substrates. We envision this effort would significantly increase the difficulty of safeguard strain decoding. One key limitation of our biocontainment strategy is the cost of the unAA supplementation for large-scale fermentation. According to our finding that 0.2 mM OMeY is required for the optimal growth of multiplex safeguard strains (Fig. S9), the extra cost of fermentation medium is estimated at $54 per

ton[58]. Despite this additional cost is acceptable for biosynthesis of high value-added products, it will be highly desirable to develop safeguard strain that exhibit robust growth at low concentration of unAA, preferably micromolar. Thus, we envision that our biocontainment of unAA-dependent yeast strains would benefit from future studies to improve the efficiency of unAA incorporation by comprehensively optimizing the translational machinery including the orthogonal aaRS/tRNA pair[59–61].

The synthetic *E. coli* with the reprogrammed genetic code provides the unique opportunity to establish the firewall from natural ecosystems[14,19,62–64]. The completion of the final Sc2.0 strain, in which all known TAG codons are substituted by TAA via genome synthesis, will pave the way for reassignment of the TAG stop codon to unAA. Our efforts and relevant knowledge obtained in this work may provide the basis to develop a robust yeast biocontainment system dependent on unAA in the upcoming final Sc2.0 strain.

## Methods
### Plasmids and oligonucleotides
All plasmid and oligonucleotides used for DNA amplification and site-directed mutagenesis in this study are listed in Supplementary Tables 3 and 4, respectively. All plasmid constructs were verified by Sanger sequencing. All essential genes driven by native promoters were cloned into centromere-based plasmids by Gibson assembly in this study. Site-directed mutagenesis was carried out to create the plasmid expressing essential gene variants with an in-frame TAG codon. *CDC27* or *NMD3* gene variants were cloned into pRS416-based vectors, and *CDC4* gene variants were cloned into pRS413-based vectors. For inducible expression of SpCas9-NG, we replaced its original promoter with the *GAL1* promoter. The synthesized cassette containing gRNA1, tRNA[Gly], gRNA2, and donor DNA was assembled with gene encoding SpCas9-NG in the pRS323-based vector. Primers (Cat number: LS-PS-00005) and synthesized DNA fragments (Cat number: LS-GS-00001) were purchased from GCATbio Co. Ltd.

### Media and culture conditions
*E. coli* DH5α was used for plasmid construction and was grown in Luria-Bertani (LB) medium in the presence of 100 μg/ml Carbenicillin (BBI Life Sciences, Cat number: A600469-0005) at 37 °C and 200 rpm. The yeast safeguard strains in this study were derived from BY4741 (*MATa his3Δ1 leu2Δ0 ura3Δ0 met15Δ0*). For plasmid maintenance, yeast strains expressing the auxotrophic marker were grown in the synthetic complete (SC) medium with corresponding amino acid dropout supplements. Temperature-sensitive (ts) mutants used to test the permissive sites for OMeY substitution were grown in medium with and without 1 mM OMeY (Aladdin, Cat number: M117085) at 20 and 37 °C. The Cdc4- and Cdc27-based synthetic auxotrophs expressing CRISPR-Cas9 system (pXF469) or control vector (pRS323 or pXF500) were grown at 30 °C in SC−Leu−His medium supplemented with 2% (w/v) galactose and 1 mM OMeY. Multiplex safeguard strains were grown in SC−Leu medium containing 2% (w/v) galactose and 1 mM OMeY at 30 °C.

### Conservation analysis of essential proteins for OMeY substitution
To identify permissive sites in essential genes for TAG codon introduction, we performed the analysis of conserved residues across all 1040 essential proteins in yeast using the database created by a previous study[65]. All these protein sequences were compared against their specific orthologs from the BLAST using the cluster-nr database (cut-off for the similarity of the ortholog was set at 0.01)[66]. The conservation visualization was generated using the Panorama viewer available on the BLAST website. The residues in the N-terminus of essential proteins that exhibit low conservation were chosen for OMeY substitution.

## Construction and modification of strains

All strains used in this study are listed in Supplementary Table 5. For construction of the ts mutants, we employed *KanMX* gene-assisted homologous recombination to introduce the target amino acid substitution in BY4741. To construct OMeY-dependent auxotrophs, BY4741 strains expressing pXF231 were co-transformed with two PCR products (essential gene fragment with an in-frame TAG codon, and *URA3* cassette), and were plated on SC–Leu–Ura medium plates with 1 mM OMeY. The transformants exhibiting OMeY-dependent growth were selected by replica-plating onto SC–Leu plates and liquid-culturing with and without 1 mM OMeY at 30 °C for 2–3 days. The same procedure was used to construct strains with two TAG codon substitutions, beginning with strains that already had one of two gene variants with an in-frame TAG codon. To construct the transcriptional-regulated safeguard strains, PCR products containing *KanMX* gene cassette, *GAL1* promoter, and the 40–60 bp upstream and 40–60 bp downstream homologous sequence were transferred into the BY4741 to replace the native promoter of selected essential genes (*RPC11*, *SKP1*, and *RPS3*). The galactose-dependent growth was verified by both liquid culturing and replica-plating. For the construction of safeguard strains to produce geraniol, overlap extension PCR was performed to generate DNA fragments containing 20 bp left homology arm, *CrGES* (GenBank: AFD64744.1), *HIS3* marker, and 320 bp right homology. Then the DNA fragments cassette was transformed into yXF307 and yXF215 for *HIS3*-based homologous recombination at the *YPL062W* gene loci. To construct the strains encoding Nmd3–GFP fusion proteins (both wild-type and NMD$_{TAG13}$), both wild-type and NMD3$_{TAG13}$ strains were co-transformed with two PCR fragments respectively (*NMD3* or *NMD3$_{TAG13}$* gene fused with *GFP* gene and *KanMX* gene cassette) to select for modified strains with G418 resistance.

## Validation of OMeY-dependent auxotroph using ts mutants

The CDC27$^{ts}$ or NMD3$^{ts}$ strain were simultaneously transformed with two plasmids expressing the LeuOmeRS/tRNA$_{CUA}$ pair (pXF231) and pRS415-based plasmids containing *CDC27* or *NMD3* gene variants, and transformants were grown on SC–Leu–Ura medium plates at 20 °C for 5–8 days. Similarly, the CDC4$^{ts}$ strain was co-transformed with pXF231 and pRS415-based plasmid carrying the *CDC4* gene variants, and transformants were grown on SC–Leu–His medium at 20 °C for 5 days. The growth of representative ts mutants for different permissive residues of selected essential genes was monitored on medium plates with and without 1 mM OMeY at both 20 and 37 °C for 3–8 days.

## Spotting assay

Isolated colonies were picked into 96-well plates and cultured overnight at 30 °C in selective liquid medium. Strains were harvested by centrifugation (3500 × *g*), washed thrice with sterile water, and re-suspended in sterile water. Ten-fold serial dilutions of the OD 1 or 0.1 yeast cells to OD 10$^{-4}$ or 10$^{-5}$ were prepared and 10 µl of these samples were spotted onto selective medium plates with and without 1 mM OMeY at 30 °C for 2–3 days. For transcriptional-regulated safeguard strains, 2% (w/v) galactose was supplemented as indicated in figure legends. Spotting assay of ts strains was conducted at 20 °C and 37 °C, respectively.

## Doubling time analysis

Kinetic growth of yeast cells was carried out using a Bioscreen C system (Lab Systems Helsinki, Finland) measured at optical density at 600 nm (OD$_{600}$). Generally, six replicates were grown in 200 µl of different media for 2 days, and blank mediums were used as the control. Then the cultures were diluted to the initial OD$_{600}$ = 0.01 with fresh medium in the microplates. Bioscreen C system was programmed to run continuously for 48 h at 30 °C and 200 rpm, and samples were taken at 20-min intervals. For measurement of safeguard strains growth in the fermentation condition, three replicates were grown in 3 ml of SC–leu medium containing 1 mM OMeY and 2% (wt/vol) galactose for overnight. Each culture was diluted to the initial OD$_{600}$ = 0.1 with 80 ml fresh medium in the Erlenmeyer flask (250 ml volume) and incubated with shanking (220 rpm) for 48 h at 30 °C. The samples were taken at 15-min intervals during the log growth phase. The OD$_{600}$ was measured manually using the cell density meter (Biochrom, Ultrospec 10). Strain doubling times were calculated as previously described[67].

## Escape assays

The individual auxotrophs were picked in biological triplicates for each strain and cultivated overnight in the corresponding medium at 30 °C, and sub-cultured into fresh permissive medium (OD$_{600}$ = 0.1) as the inoculum. Two technical repetitions were conducted for each colony by dividing the inoculum into two separate inocula grown to the late-log phase (OD$_{600}$ = 4–6) at 30 °C with shaking at 200 rpm. The cultures were centrifuged at 3500 × *g* for 5 min, washed thrice with sterile water, and re-suspended in sterile water before plating. Generally, the escape frequency was calculated by dividing the number of escapees grown on the non-permissive plates by the total number of CFUs cultivated for plating. Based on the commonly accepted point that one OD$_{600}$ is equivalent to around 10$^7$ yeast CFUs/ml[68], we performed five serial 10-fold dilutions and plated 100 µl of the diluted sample (OD$_{600}$ = 0.001) on the permissive solid medium plates (90 mm diameter) to determine viable titer for each experimental group, which often resulted in around 600 to 1000 CFUs. The actual observed CFUs allowed us to determine the conversion factor used to convert between the number of CFUs cultivated and OD$_{600}$. This conversion factor was utilized to normalize the total number of viable yeast cells plated on non-permissive plate for calculating the escape frequency. Around 10$^7$–10$^9$ cells determined by OD$_{600}$ in the permissive cultures were plated and spread using glass beads onto non-permissive solid medium plates (150 mm diameter), which were then incubated at 30 °C for 14 days. The detection limit of escape frequency is below ~1.0 × 10$^{-9}$ escapees/CFUs. The suspect escapees were further repatched on non-permissive solid medium plates for monitoring to confirm whether they were indeed escape mutants.

## Assay of employing CRISPR/Cas9 system to prevent the occurrence of amber suppressor tRNA$^{Tyr}$

Two OMeY-dependent strains (yXF223 and yXF221) were separately transformed with CRISPR/Cas9 plasmid (pXF469), empty vector pRS423, or control plasmid only containing SpCas9-NG but not gRNAs (pXF500). Isolated single colonies of transformants were cultivated at 30 °C in SC–Leu–His medium supplement with 1 mM OMeY and 2% (w/v) galactose to induce SpCas9-NG gene expression. The cell cultures were collected at different intervals from day 3 to 12 and the escape frequency was measured using the same method as described before. To confirm the editing activity of CRISPR-Cas9 system, the escapees derived from OMeY-dependent strains expressing pXF469 were analyzed by whole genome sequencing.

## Fitness-oriented screening for permissive residues at N-terminus

To construct a library of DNA fragments targeting amino acid substitution at the N-terminus (from residue 4 to 20), the synthesized DNA fragments containing the native promoter of *CDC27* or *CDC4* gene (150 bp) and the coding region (163 bp) with an in-frame TAG substitution were mixed with equal moles and were assembled with the auxotrophic marker *URA3* by overlap extension PCR. Primers used in the library construction are listed in Supplementary Table 4. Around 1000 ng PCR product of the library was transformed into the multiplex safeguard strain yFX241, the transformant was plated on SC–Leu–Ura medium supplement with 1 mM OMeY and 2% (w/v) galactose at 30 °C for 3 days. The randomly selected transformants were swirled into 96-

well plates with 200 µl sterile water, and were then plated on the permissive and non-permissive solid medium using ROTOR pining robot (Singer Instruments) to select candidate OMeY-dependent auxotrophs. The corresponding gene fragments containing the TAG substitution site of all selected auxotrophs were amplified by colony PCR and then verified using Sanger sequencing.

## GFP-based fluorescence assay

Isolated colonies of strains encoding Nmd3–GFP fusion proteins (both wild-type and NMD$_{TAG13}$) were cultivated in 1 ml of SC–Leu–Ura medium (initial OD$_{600}$ = 0.1) with 1 mM OMeY and grown to stationary phase at 30 °C for 24 h. The cultures were centrifuged at 3500 × $g$ for 3 min, washed thrice with sterile water, and re-suspended by 1 ml sterile water. OD$_{600}$ and green fluorescence ($\lambda_{ex}$ = 480 nm, $\lambda_{em}$ = 525 nm) were measured using a microplate reader (Tecan Spark) using 200 µl of resuspending in a flat black 96-well plate. The relative fluorescence intensity per unit biomass was calculated as fluorescence intensity/OD$_{600}$.

## Measurement of dynamic change of safeguard strains in fed-batch fermentation

For the first round of fermentation, a single colony of safeguard strain yXF281 was inoculated into 5 ml of SC–Leu medium supplement with 1 mM OMeY and 2% (w/v) galactose at 30 °C for 12 h. The seed culture was transferred into 150 ml of fresh medium with an initial OD$_{600}$ of 0.1, cultivated at 30 °C with shaking at 200 rpm, harvested in the log phase, and then inoculated into a 3-liter fermenter (Applikon Biotechnology, the Netherlands). The fermenter working volume was 1 liter. Galactose was used as the carbon source in a fed-batch fermentation at 30 °C. The pH was automatically maintained at 6.0 using 2 M NaOH, the airflow rate was set at 1 vvm, and the rate of agitation was set at 220 rpm. After 48 h of fermentation, galactose concentration in the medium was controlled at a flow rate of 8 ml/h by feeding 500 g/l galactose. Ten ml of culture was sampled to quantify strain density (OD$_{600}$) and titers of geraniol at 72 h of fermentation. In the second round of fermentation, fermenter was washed with 3-liter sterilized water before inoculation of seed culture of strain yXF282. Fermentation was carried out in SC–Leu medium without OMeY, and glucose was used as the carbon source. After 48 h of fermentation, 500 g/l glucose was fed to the fermenter by controlling feed rate similar to galactose. The culture was sampled at different timepoints during the second round of fermentation to determine the CFU number of total viable cells in the fermentation broth (yXF281 and yXF282) and remaining safeguard strain (yXF281). Viable CFU of total cells was determined by using ten-fold serial dilutions on SC–Leu solid medium supplement with 1 mM OMeY and 2% (w/v) galactose for 3 days. As yXF281 carries the $URA3$ and $KanMX$ genes, viable CFU of yXF281 was determined by using ten-fold serial dilutions of $10^4$–$10^9$ cells on SC–Leu–Ura selective medium plates supplement with 1 mM OMeY, 2% (w/v) galactose, and G418 (200 µg/ml) for 3 days. The probability of successfully sub-culturing safeguard strain yXF281 from the fermentation broth was calculated according to Eq. (1) as shown below:

$$P = 1 - \left( \prod_{n=1}^{k} \left( 1 - \frac{1}{n} \times \min\left( \frac{V \times N}{k}, 1 \right) \right) \right) \quad (1)$$

where $k$ represents the total number of all combinations of required compounds ($k$ = 60,000 in this study), $V$ is the assumed volume (ml) used for sub-culturing of safeguard strains in a real fermentation scenario ($V$ = 1 ml in this study), and $N$ is the viable CFU per ml.

## Whole genome sequencing analysis of the escapees

Single colony of escape mutants were randomly picked from different non-permissive plates. The escapees were grown in YPD medium

containing 2% (w/v) glucose, 10 g/l yeast extract, and 20 g/l peptone at a temperature of 30 °C until stationary phase. Genomic DNA was extracted using the glass bead method after cell lysis. High-throughput sequencing was carried out on the BGISEQ-T7 platform following established protocols. Each colony generated ~4.86 Gb of data, that indicates ~400-fold genome coverage. Low-quality reads or reads with unknown bases were removed using SOAPnuke v2.1.1[69], with less than 15% of reads filtered. The raw sequenced reads were then mapped against the reference sequence of BY4741 using BWA v2.2.5[70]. The genetic mutations containing SNPs, insertions, and deletions in the genome of the escapees were identified by GATK v2.7 with default settings[71].

## Reporting summary

Further information on research design is available in the Nature Portfolio Reporting Summary linked to this article.

## Data availability

The DNA sequencing data of this study has been deposited in the NCBI database under accession code PRJNA967796 and CNSA (CNGB Nucleotide Sequence Archive) under accession number CNP0004223. The conservation analysis for all essential proteins of *S. cerevisiae* compared against their specific orthologs by BLAST uses the NCBI built-in cluster-nr database (https://blast.ncbi.nlm.nih.gov/Blast.cgi?PROGRAM=blastp&PAGE_TYPE=BlastSearch). All data supporting the findings of this study are available within the manuscript file and its Supplementary Information files. Source data are provided with this paper.

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

## Acknowledgements

This work was supported by grants from National Key Research and Development Program of China (No. 2018YFA0900100 to X.F.); Natural Science Foundation of Guangdong Province, China (No. 2021A1515010995 to X.F.); Tip-top Scientific and Technical Innovative Youth Talents of Guangdong Special Support Program (2019TQ05Y876 to Y.S.). We are grateful to Prof. James A Van Deventer (Tufts University) for sharing the LeuOmeRS/tRNA$_{CUA}$ plasmid and to Prof. Zihe Liu (Beijing University of Chemical Technology) for sharing the plasmid expressing SpCas9-NG. We also thank Prof. Junbiao Dai (Shenzhen Institutes of Advanced Technology) for sharing the ts mutant strains. We also thank the technical support provided by China National GeneBank.

## Author contributions

X.F. and Y.S. supervised the study. X.F. designed the experiments. T.C., W.D., S.Y., Y.Z., Y.H. and J.Z. performed experiments. Y.W. and Huiming Z. assisted with sequencing analysis of escapees. T.C., W.D., X.F., S.Y., Haoling Z., D.W. and Z.P. were responsible for data acquisition and analysis. W.Z. and X.X. interpreted the results. X.F. wrote the manuscript with inputs from all the other authors. Y.S. edited the manuscript. All authors commented on the final draft of the manuscript.

## Competing interests

The authors declare the following competing interests: X.F., Y.S., T.C., W.D., S.Y. and BGI-Shenzhen have filed patent applications (Chinese patent application number: 202210557275.X and PCT/CN2023/107221) describing the biocontainment methods reported in this study. The remaining authors declare no competing interests.
