## [Peer Review File · Nature Communications]

Reviewers' Comments:

Reviewer #1:

Remarks to the Author:

A. Summary of key results: *Saccharomyces cerevisiae* is an important host for industrial fermentation and is generally more difficult to demonstrate genetic code expansion in than bacterial hosts due to processes such as nonsense mediated decay. In this manuscript, the authors provide a compelling justification for implementing the intrinsic biological containment system of synthetic auxotrophy in this organism and a comprehensive description of the relevant literature. The manuscript attempts to describe a general framework for creation of synthetic auxotrophs, which considers some aspects that were not previously considered by the small number of others who demonstrated synthetic auxotrophy in *E. coli*. The authors make use of valuable tools such as a temperature sensitive mutant collection and knockouts of the non-sense mediated decay pathway. The authors also characterize the observed escapees (or escapers) and test an innovative approach to minimize the appearance of amber suppressor tRNAs using CRISPR/Cas9. The authors then demonstrate multi-layered biocontainment using synthetic auxotrophy and galactose-dependent transcriptional control. They then concluded their study with an analysis of a real fermentation, which will be commented on later below.

B. Originality and significance: Overall, there are many interesting and new insights from this study, and the topic is of high importance. Biological containment and biosecurity are especially heightened topics of discussion after the COVID-19 pandemic, if not for preventing espionage of industrial strains then certainly at least for preventing accidental release of microorganisms that could contain recombinant DNA of concern. As mentioned above, this article features original content in multiple directions, including in the approach towards determining suitable synthetic auxotrophic markers and sites for unAA incorporation as well as the system for immunity to amber suppressor tRNAs. Finally, yeast is an important strain and this work synergizes with other ongoing high profile work in the community, such as the Sc2.0 genome synthesis project.

C. Data and methodology: Here, the authors could improve their work in certain aspects. They should be commended for showing their serial dilutions, which build reader confidence in their estimates of total colony forming units observed on permissive media. However, the reported assay detection limit of 1×10^{-9} escapees per CFU is suspect as it is just one detection limit listed for all experiments throughout the manuscript. Detection limits for escape assays are based on the total number of CFU cultivated and analyzed for a given strain. As it is not possible to cultivate exactly the same number of cells in different cultures, the assay detection limit should be distinct for each experimental group and it should preferably be an average of three replicates. When escape frequencies are several orders of magnitude above the detection limit (such as the 10^{-4} to 10^{-7} range), then this point is not so important. But it is rather critical for the cases where escape was not detected, and the authors' approach is below the standard set in the literature. This reviewer would think the authors have collected the data for total CFUs in every case, so hopefully it does not require new experiments and instead simply requires reporting more information.

D. Appropriate use of statistics and treatment of data: Statistics appear appropriate, aside from the point raised above about detection limit as well as the uncertainty about the number of replicates for certain procedures. For example, the method section for the "Escape Assays" begins by stating that auxotrophs were picked in biological triplicates, but are the colony counts from plating on permissive or non-permissive solid-media done once for each strain or in triplicate for each strain? As both escape frequencies and serial dilutions tend to exhibit high variance, it would be ideal if the CFU estimates were conducted in triplicate for each strain. This aspect becomes further confusing as the figure captions refer to the mean and standard error of the mean of six biological replicates rather than three. Does that mean for each of the three biological replicates the CFU estimation was conducted in duplicate?

E. Conclusion: Most of the conclusions presented in the paper are supported by the data, with some exceptions regarding the fitness of the synthetic auxotrophs and the real fermentation scenario. I will comment on the real fermentation experiment later. Regarding the fitness of the synthetic auxotrophs, it is difficult to draw conclusions based on how the doubling time data is represented. Could they be represented as fold changes relative to the wild-type growth? Is there a statistically significant difference in the WT doubling time and the nsAA-dependent strain doubling time? How much worse is the doubling time? How does this difference scale to fermentation? For 4d in line 280, the authors suggest that there are not too many fitness impacts,

but then walk back that statement shortly after (line 290). Were the authors referring to the doubling time data in Fig 2 since they do not test the strains containing the plasmid for CRISPR disruption in the fitness testing in Fig 5? Could the y-axis be rescaled or focus in on the doubling times of interest to better show the differences between strains? In general, the text makes several strong claims about certain strains having better fitness than others, but the data does not necessarily show that.

Major Comments

- The manuscript text references SI figures (e.g. Fig. S4), but the SI document lists Extended Data Figures and no Fig. S#. This may be a simple swap of nomenclature, but many Nature family submissions have both extended data figures and supplementary figures so it is important to be consistent.
- The strain table and primer table are quite well prepared and informative – this reviewer particularly appreciates the “description and construct generated” column in the primer table. In contrast, the plasmid table seems much less informative and is very difficult to follow. The first row entry references a pRS315 – is that a standard plasmid that is commercially available, and is its sequence publicly available? The next entry references pRS416 – same questions for this starting plasmid, not to mention the actual plasmids created in this study? It is valuable to the community and important for reproducibility if the authors can deposit this sequence information onto GenBank/NCBI – for example, so that the exact promoter sequences used are known. The authors should be commended for their deposition of the raw sequence reads, though that information is not as simple to sort through as a well-annotated plasmid sequence.
- The real fermentation experiment is described in a confusing manner in both the text and Figure 6 – to the point where a reader is unlikely to understand how it was conducted. From Line 323 onward in that paragraph, it is not clear at all what the authors did in this experiment. The figure suggests that the fermentation began as one biocontained strain that was modified to produce geraniol, but that after washing cells a second strain was added. Why was there only 8000 cfu of biocontained strain per ml at this point, after growth in what should have been permissive media? How is the inoculation of a fermentation with a second strain that does not contribute to the fermentation representative of a “real fermentation”? Is its only role to outcompete the biocontained strain in the non-permissive media condition? What was the duration of fermentation with the biocontained strain only prior to switching growth conditions from permissive to non-permissive?
- The authors have based their system on the LeuOmeRS/tRNACUA system derived from the EcLeuRS for incorporation of O-methyl-L-tyrosine (OMeY). First, since the unAA is OMeY, is the name of the aminoacyl-tRNA synthetase OMeYRS instead of OmeRS? Second, the fidelity of the orthogonal translation systems are always a concern for synthetic auxotrophy applications, especially when the essential proteins are not computationally designed to accommodate only an unAA. This is a highly pertinent question as it seems the authors have consistently used a Leucine dropout media for their experiments. Their use of the Leu marker for cloning justifies the use of a Leucine dropout media for strain construction but not for escape assays, particularly as this manuscript attempts to describe a real fermentation scenario. In real fermentation or the environment, leucine or tyrosine (structurally similar to OMeY) may be present in higher abundance. The authors should conduct experiments with at least the strains that are not reported to escape (or the strains that have low detectable escape rates) to show to readers whether the system breaks in the presence of 1-5 mM Leucine or 1-5 mM tyrosine. In my opinion, if marginally higher escape rates are detected then this is not a major barrier to publication in this journal given how well the system works in the dropout condition, but it is vital to share how context-dependent the observed behavior is in regard to amino acid composition.
- When coupling synthetic auxotrophy to the galactose kill switch, the authors should provide more explanation of the intended uses or limitations of the kill switch. It does not seem like a particularly useful technology because it leads to cell death if glucose is around which is not a great trait to have, either for a real fermentation (given how glucose is usually the desired carbon source) or for accidental release.

Minor Comments

- In the introduction, the authors should briefly clarify that the complete Sc2.0 strain has not yet been created but that biocontainment is an important goal for that strain.
- line 47: when first describing the GMO escapee rate, the authors should specify that this is the maximum acceptable rate of escape from intrinsic biological containment, and that the units

presented are escapees per colony forming unit.

- Lines 95-98: When the authors revise their description of the real fermentation experiment, they should ensure that this sentence contains more clarifying language about what they tested.
- Lines 159-161: The authors claim that there is no growth without OMeY but then escape rates are provided (10^{-5} - 10^{-8}). The same issue arises later for Fig 2b. Could the authors clarify what they mean by no growth in these cases? Also, the non-diluted samples in the serial dilution images appear faintly. Is this a film reflective of outgrowth that never becomes more dense, or do the strains escape when plated at high density? It would be helpful if the authors specify at what time the images of the serial dilutions are taken and how long the observation period is for these cases, because it may be the case that at high density of plating the escape rates are different.
- Lines 174: Could the authors provide more explanation of how they chose these 4 genes?
- Lines 166, 177: When values of escape or escape rates are provided, they should be followed by these units (escapees/CFU or escapers/CFU). Escapees would be preferable language based on literature precedent.
- Line 117: It would be helpful in this paragraph to inform the reader what the essential functions of CDC4 and CDC27 are, and why they could not be complemented by cross-feeding of metabolites.
- Line 173: "series"
- Line 181: "The ts strain-based spotting assay suggested OMeY incorporation efficiency by LeuOmeRS/tRNACUA might not be enough to produce sufficient amount of Nmd3 protein for cell survival (Fig. S4)." This seems like the most likely reason why cell survival might not occur, but the spotting assay does not directly indicate anything about the amount of Nmd3 protein being produced. If the authors wish to make this statement they would need to use a different kind of assay technique. Instead, the authors could say that "The spotting assay indicated that cell survival in permissive media at high temperatures was only faintly observed in conditions of limited or no dilution (Fig. S4), which is likely because insufficient Nmd3 protein was produced."
- When escape was observed from the temperature sensitive mutant screen, could the authors shed more light on why that may be occurring?
- Line 186: The claim about transcriptional rates seems difficult to justify when a gene corresponding to only one high RKPM value (300) did not function well but the gene with the next highest RKPM value (194) did function well. Either the claim should be softened or genes with more RKPM values in the 200-350 range should be tested to see if this is a real trend rather than one unique case.
- Line 190: More description around this section about how the escape assay was conducted may help readers newer to this concept.
- Line 212: Please clarify that this improved fitness is in non-permissive conditions.
- Line 243: In the text it would be helpful if the authors could list the escape rates obtained with and without the use of the CRISPR-Cas9 system, or at least representative or best values.
- Line 249: it wasn't clear what these leftover scar interrupts were. are these mutations that did not disrupt the PAM sequence but that altered the adjacent region to create mismatches to the guide RNA?
- Line 259: How did the authors settle on these genes?
- Line 280: The conclusion for the doubling time data here is inconsistent with the data observed
- Lines 287-288- What was the motivation here if, in the previous paragraph, the authors say there was not fitness impairment?
- Line 300: "revealed"
- Line 330: Avoid the word "significantly" unless stating statistical significance
- Line 333: Could the authors state what G418 is? Seems like an antibiotic but it is unclear.
- Line 371: Revise to make clearer that this statement is about the past, before this study: "The escape mechanisms that disable the unAA-dependent biocontainment system in *S. cerevisiae* were previously unexplored."
- Lines 402-404: The authors' perspective on the difficulty of identifying how to grow a synthetic auxotroph would be far better supported if they could show that their synthetic auxotrophs are not capable of growing on other unAAs. It is widely known that many orthogonal translation systems exhibit polyspecificity. Earlier, this reviewer pointed out that the growth on media containing Leu or supplemented Tyr should be tested. For claims such as these, the authors should consider testing other unAAs, or they should temper their remarks.
- Line 443: typo for "safeguard"
- Line 455: It is unclear how the analysis of conserved residues was performed and the

subsequent data representation in the extended data figures. Was the conservation analysis performed across all essential proteins in yeast? Were those genes compared against specific orthologs from the BLAST search? Was there a cutoff for the similarity of the ortholog?

- Line 476: typo for "varied"
- Line 518: typo for "medium"
- Line 520: Please clarify the suspected dilution for plating on permissive media? What is the precedence for approximating CFU counts with this method?
- Line 584: the number for k seems to be incorrect, or at least the comma is placed at the wrong location
- I may have missed this, but when two TAG codons are used together in the same gene, or when the transcriptional and translational control systems are used together, what are the mechanisms of escape when escape is observed?
- Figure 1: the doubling time plot in panel C does not have the bar graphs labeled on the x-axis. one would think that maybe the labels on the bottom plot in panel C could apply to the top plot, but there are 10 bars in the top plot and only 9 strains shown on the bottom plot. my guess is that the first 9 bars correspond to these strains in the same order, and that the last bar is the WT strain, but I should not have to guess. please provide labels in the top plot.
- (this is just a comment, no corrective action is required) Figure 2: while the serial dilutions shown in 1b, 2a, and throughout the SI do not provide much quantitative information, I appreciate their inclusion as it shows the reliability of the serial dilutions. Presumably, larger volumes of the same serial dilution samples were used for the measurement of escape frequency.
- Figure 2: bottom plot in 2b should state that this was escape frequency seen at 2 days, independent of whether figure 1 states this or the figure caption states this
- Figure 3: In figure 3a, it is unclear why the nearly identical image is shown twice - the "no cleavage", "normal strains", "gene encoding tRNATyr". The only difference is the inclusion or not of Cas9 and guide, but it feels like the information is redundant. The concept could be better illustrated by showing the wild-type strain and sequence first with this gRNA and SpCas9-NG unbound to the wild-type sequence, and to then have an arrow showing that if an escaper arises within the population then the image that is currently the first image on the left would apply. Note that this image is currently somewhat redundant with the third image in this series, so if the authors were to follow these suggestions then they would simplify the image from 4 to 2.
- Figure 4: Figure 4b confuses me. According to Fig. 4a and the manuscript text, the 2% glucose condition should be non-permissive and the 2% galactose condition permissive, correct? If that is the case then Figure 4b has this labeled backwards. Also, the depiction of serial dilution for the transcriptional kill switch raises the question of what the escape rates were since this does not seem to work as well as synthetic auxotrophy. that is consistent with past literature but would be good to reinforce. were the escape rate measured?
- Figure 6: As mentioned in the Major Comments section, I strongly suggest revising this figure to provide more conceptual depiction of the experiment. I also think the inclusion of the probabilities above the last two bars is distracting and unnecessary – these values can simply be referenced in the text.
- When the authors comment on a real fermentation scenario, they should provide readers with a better sense of the total CFU expected of yeast cells at different scales of fermentation, and they should also provide some justification for why they have chosen to culture only $\sim 10^9$ cells since prior studies have reported lower assay detection limits – in many cases 2 or 3 orders of magnitude lower. Practically speaking, plating 10^9 cells is more convenient, especially if the goal is to be below the NIH biocontainment standard. However, for real fermentation one would want to know if the strains that do not exhibit observable escape in this study are actually suitable or if they simply escape at 10^{-10} escapees per CFU.

Reviewer #2:

Remarks to the Author:

Chang et al. presented a novel two-layered approach for the construction of a biocontainment system. The first layer involves the implementation of an unnatural amino acid dependent gene expression system by incorporating an amber codon at the permissive site of an essential gene. The second layer utilizes a transcriptional control mechanism through the installation of a galactose-inducible promoter, enabling the expression of another essential gene.

In addition to leveraging a well-established amber suppression strategy previously employed in bacterial cells, the authors developed a galactose-inducible gene expression system to enhance the stringency of the biocontainment system in eukaryotic cells.

While the manuscript is generally well-written, I would suggest revising the following aspects:

1. [Page 5, line 122] References 15 and 16, which are also related to the "design" approach, should be included alongside the other references.
2. [Pages 5 and 6, lines 121–133] [Pages 13 and 14, lines 364–367] The strategy of identifying permissive sites has been extensively studied in previous research (ref. 12, Rovner et al., Nature 2015). The authors should compare the two algorithms and explain the advantages of the current approach over the previous one.
3. [Page 7, line 173] "serious" → "series"
4. [Page 7, lines 180–183] It is necessary to clarify whether the mutation site of NMD3 is permissive. Additionally, what is the minimal expression level of NMD3? Overall, more experiments are needed to confirm whether the inability to obtain an OMeY auxotroph by targeting NMD3 is caused by an insufficient amount of Nmd3 protein. For example, determining the minimal expression level of Nmd3 for cell survival, testing the suppression efficiency of the LeuOMeRS/tRNA system, and screening for permissive sites of Nmd3.
5. [Page 7, lines 185–186] The statement "has good versatility for many different essential proteins in *S. cerevisiae*" is exaggerated, considering that it only includes the group of low-expression genes.
6. [Page 8, line 201] [Page 14, line 374] "Si, L. et al. (2016) Generation of influenza A viruses as live but replication-incompetent virus vaccines. Science, 354, 1170–1173." should be included in the references.
7. [Page 8, line 206] The authors should discuss other possibilities of misincorporation of natural amino acids by LeuOmeRS/tRNA in the absence of OMeY.
8. [Page 9, lines 221–222] While the CRISPR-Cas9 strategy is a good idea, it had little impact on the biocontainment system, possibly due to its limited targeting on tRNA-Tyr, not other escape mechanisms. This limitation should be discussed, along with future directions for this "immunity" based approach.
9. [Page 10, lines 267, and Figure 4b] The images for galactose and glucose conditions appear to be swapped.
10. [Page 12, line 307] "reside" → "residues"; "residue 7,8,11,14" → "residues 7, 8, 11, and 14"
11. [Page 12, lines 335] While it is true that more than 200 unnatural amino acids can be genetically encoded, it should be confirmed how many of these can be encoded in yeast, as most host organisms are bacteria, not yeast.
12. The transcriptional-based switch does not seem to be a perfect biocontainment system because unlike unnatural amino acids, galactose is naturally available. The authors should include a discussion regarding this limitation.

We would like to thank the reviewers for their time and suggestions that helped us improve our work. Below, we provide a point-by-point response (in blue color) to reviewers' comments.

Reviewer #1 (Remarks to the Author):

A. Summary of key results: *Saccharomyces cerevisiae* is an important host for industrial fermentation and is generally more difficult to demonstrate genetic code expansion in than bacterial hosts due to processes such as nonsense mediated decay. In this manuscript, the authors provide a compelling justification for implementing the intrinsic biological containment system of synthetic auxotrophy in this organism and a comprehensive description of the relevant literature. The manuscript attempts to describe a general framework for creation of synthetic auxotrophs, which considers some aspects that were not previously considered by the small number of others who demonstrated synthetic auxotrophy in *E. coli*. The authors make use of valuable tools such as a temperature sensitive mutant collection and knockouts of the non-sense mediated decay pathway. The authors also characterize the observed escapees (or escapers) and test an innovative approach to minimize the appearance of amber suppressor tRNAs using CRISPR/Cas9. The authors then demonstrate multi-layered biocontainment using synthetic auxotrophy and galactose-dependent transcriptional control. They then concluded their study with an analysis of a real fermentation, which will be commented on later below.

Response 1: We thank the reviewer for in-depth and comprehensive interpretation of our work.

B. Originality and significance: Overall, there are many interesting and new insights from this study, and the topic is of high importance. Biological containment and biosecurity are especially heightened topics of discussion after the COVID-19 pandemic, if not for preventing espionage of industrial strains then certainly at least for preventing accidental release of microorganisms that could contain recombinant DNA of concern. As mentioned above, this article features original content in multiple directions, including in the approach towards determining suitable synthetic auxotrophic markers and sites for unAA incorporation as well as the system for immunity to amber suppressor tRNAs. Finally, yeast is an important strain and this work synergizes with other ongoing high profile work in the community, such as the Sc2.0 genome synthesis project.

Response 2: We thank the reviewer for recognizing the values and originality of our study.

C. Data and methodology: Here, the authors could improve their work in certain aspects. They should be commended for showing their serial dilutions, which build reader confidence in their estimates of total colony forming units observed on

permissive media. However, the reported assay detection limit of 1×10^{-9} escapees per CFU is suspect as it is just one detection limit listed for all experiments throughout the manuscript. Detection limits for escape assays are based on the total number of CFU cultivated and analyzed for a given strain. As it is not possible to cultivate exactly the same number of cells in different cultures, the assay detection limit should be distinct for each experimental group and it should preferably be an average of three replicates. When escape frequencies are several orders of magnitude above the detection limit (such as the 10^{-4} to 10^{-7} range), then this point is not so important. But it is rather critical for the cases where escape was not detected, and the authors' approach is below the standard set in the literature. This reviewer would think the authors have collected the data for total CFUs in every case, so hopefully it does not require new experiments and instead simply requires reporting more information.

Response 3: We thank the reviewer for providing helpful comments and suggestions to improve our work. In the revised manuscript, we provide the Source Data file that shows the raw data used to calculate the escape frequency, including the total cultivated CFUs for plating and CFUs of escapers grown on the non-permissive plates in every case. Specifically, the total number of CFUs cultivated for plating is determined by the OD_{600} of cell culture, the volume of cell culture used for plating and the conversion factor used to convert between the number of CFU cultivated and OD_{600} for each strain. For each experimental group, we performed five serial dilutions at 10-fold and plated 100 μ L of diluted sample ($OD_{600}=0.001$) on the permissive solid medium to determine viable titer. As it is known that one OD_{600} unit of budding yeast culture is equivalent to around 1×10^7 CFUs/mL (PMID: 14970646, 36707648), plating 100 μ L of the serial diluted sample ($OD_{600}=0.001$) would expect to result in ~ 1000 CFUs. Consistent with previous findings, we indeed obtained around 600-1000 CFUs after plating for most of experimental groups (included in the source data), meaning one OD_{600} unit measured by us is equivalent to $0.6-1.0 \times 10^7$ CFUs/mL. Based on these data, we could then determine the conversion factor used to convert between the number of CFU cultivated and OD_{600} for each case. We used different conversion factors to normalize the total number of viable yeast cells for plating on non-permissive plate and for calculating the escape frequency. We also performed an additional experiment using a wild-type strain to demonstrate the reliability of our serial dilutions and OD_{600} measurement by our spectrophotometer. As shown in the figure below, we observed a strong correlation between CFUs and serial dilutions. In addition, ~ 900 CFUs were detected after plating 100 μ L of the diluted wild-type strain ($OD_{600}=0.001$), which falls in the same range of plating 100 μ L of diluted synthetic auxotrophs ($OD_{600}=0.001$). Thus, all these data suggest our estimate of total CFUs observed on permissive media is solid.

We agree with the reviewer “it is not possible to cultivate exactly the same number of cells in different cultures”. Indeed, the assay detection limit is slightly distinct (but very close or equal to 10^9 CFUs) for different experimental groups. For clarity, we now specifically add the description of the actual assay limit (exact cultivated CFUs for plating), in the updated figure legend for experimental groups that we did not detect any escapers.

Our choice of $\sim 1 \times 10^{-9}$ escapees per CFUs as the detection limit is based on the consideration of reasonable amount of works and relevant peer-reviewed literatures. In our escape assays, serial dilutions of 10^7 - 10^9 cultivated CFUs were plated on non-permissive solid medium plates (150 mm diameter). Please note that plating of 10^9 CFUs requires the preparation of around 50 ml cell cultures (late-log phase, $OD_{600}=3$) and use of five non-permissive plates (150 mm diameter). We performed six replicates (three biological replicates which was conducted in duplicate) for each strain. Therefore, preparation of $\sim 50 \times 6 = 300$ mL cell culture and plating of $5 \times 6=30$ big plates were needed just for one group aiming at the 10^9 CFUs. If we set the detection limit as 1×10^{-10} , the amount of experiment works would be exceptionally high throughout this study. In addition, please also note that a recent Nat Commun paper (PMID: 33122649) that developed yeast biocontainment systems dependent on fluoride sensitivity also chose the same detection limit (1×10^{-9}) as our work. For another PNAS paper study that developed transcriptional safeguard strains (PMID: 28174266), the maximal 10^8 yeast cells were used to determine escape rate. Thus, we

think our assay limit is reasonable and reaches the very high standard set according to peer-reviewed articles about yeast-based biocontainment system.

D. Appropriate use of statistics and treatment of data: Statistics appear appropriate, aside from the point raised above about detection limit as well as the uncertainty about the number of replicates for certain procedures. For example, the method section for the “Escape Assays” begins by stating that auxotrophs were picked in biological triplicates, but are the colony counts from plating on permissive or non-permissive solid-media done once for each strain or in triplicate for each strain? As both escape frequencies and serial dilutions tend to exhibit high variance, it would be ideal if the CFU estimates were conducted in triplicate for each strain. This aspect becomes further confusing as the figure captions refer to the mean and standard error of the mean of six biological replicates rather than three. Does that mean for each of the three biological replicates the CFU estimation was conducted in duplicate?

Response 4: We apology for the confusion involving the number of replicates for certain procedures. As mentioned by the reviewer, three biological replicates (distinct colonies after transformation) were used for each strain, and each of the three biological replicates was conducted in duplicate (two technical repetitions). To avoid confusion about the number of replicates, we rephrased the relevant sentences in the updated Figure legend and method describing “Escape assays”. For the issue of high variance observed in certain groups, please note that the variance between the two technical repetitions was found to be minor in most cases (see Source Data file for details). We also repeat experiments to replace the data (e.g. ERG8 in Fig. 2b) that show high variance, and newly obtained data shows small variance among different replicates.

E. Conclusion: Most of the conclusions presented in the paper are supported by the data, with some exceptions regarding the fitness of the synthetic auxotrophs and the real fermentation scenario. I will comment on the real fermentation experiment later. Regarding the fitness of the synthetic auxotrophs, it is difficult to draw conclusions based on how the doubling time data is represented. Could they be represented as fold changes relative to the wild-type growth? Is there a statistically significant difference in the WT doubling time and the nsAA-dependent strain doubling time? How much worse is the doubling time? How does this difference scale to fermentation? For 4d in line 280, the authors suggest that there are not too many fitness impacts, but then walk back that statement shortly after (line 290). Were the authors referring to the doubling time data in Fig 2 since they do not test the strains containing the plasmid for CRISPR disruption in the fitness testing in Fig 5? Could the y-axis be rescaled or focus in on the doubling times of interest to better show the differences between strains? In general, the text makes several strong claims about certain strains having better fitness than others, but the data does not necessarily show that.

Response 5: We thank the reviewer for providing helpful comments to improve our

work. To follow the reviewer's suggestions, the y-axis of doubling time data is now rescaled (starting from 1.5 hours) to better show the differences between various strains. At line 290, we refer to the doubling time difference between CDC4_{TAG9} and CDC4_{TAG325} in Fig. 4d to support our claim that "OMeY incorporation at different permissive site affects the growth rate". To avoid this confusion, we now specifically point out comparison objects when citing Fig. 4d in the updated manuscript (line 311). In addition, we now add p-value tests to determine statistical significance of doubling time between the WT strain and the unAA-dependent strains.

To answer the reviewers' question "How does this difference scale to fermentation?", we performed additional experiments to measure the growth curve of representative multiplex safeguard strains (replacing the residues 4, 8, 12, 14, 15 and 18 of Cdc27 by OMeY) with distinct growth rate that were grown in 80 ml fermentation condition. We found the fold changes of doubling time relative to the wild-type growth are highly comparable between different growth conditions (0.2 mL versus 80 mL) as shown in the figure below (please find details in Fig. S9 and revised manuscript in line 326-330).

Supplementary Figure 9. Doubling time comparison of Cdc27-based two-layered safeguard strains grown in small and large scale.

Major Comments

- The manuscript text references SI figures (e.g. Fig. S4), but the SI document lists Extended Data Figures and no Fig. S#. This may be a simple swap of nomenclature, but many Nature family submissions have both extended data figures and supplementary figures so it is important to be consistent.

Response 6: We thank the reviewer for this comment. SI document is now modified to be consistent with manuscript text.

- The strain table and primer table are quite well prepared and informative – this reviewer particularly appreciates the “description and construct generated” column in

the primer table. In contrast, the plasmid table seems much less informative and is very difficult to follow. The first row entry references a pRS315 – is that a standard plasmid that is commercially available, and is its sequence publicly available? The next entry references pRS416 – same questions for this starting plasmid, not to mention the actual plasmids created in this study? It is valuable to the community and important for reproducibility if the authors can deposit this sequence information onto GenBank/NCBI – for example, so that the exact promoter sequences used are known. The authors should be commended for their deposition of the raw sequence reads, though that information is not as simple to sort through as a well-annotated plasmid sequence.

Response 7: We thank the reviewer for this suggestion. pRS315 and pRS416 are commonly used plasmids that are commercially available from ATCC (cat. No 87521 and 77144 for pRS416 and pRS315 respectively), and their sequences are also publicly available from SnapGene website. To follow the reviewer's suggestion, we now provide the sequence information for plasmids that are constructed in this study at the end of the SI document with specific annotations.

- The real fermentation experiment is described in a confusing manner in both the text and Figure 6 – to the point where a reader is unlikely to understand how it was conducted. From Line 323 onward in that paragraph, it is not clear at all what the authors did in this experiment. The figure suggests that the fermentation began as one biocontained strain that was modified to produce geraniol, but that after washing cells a second strain was added. Why was there only 8000 cfu of biocontained strain per ml at this point, after growth in what should have been permissive media? How is the inoculation of a fermentation with a second strain that does not contribute to the fermentation representative of a “real fermentation”? is its only role to outcompete the biocontained strain in the non-permissive media condition? What was the duration of fermentation with the biocontained strain only prior to switching growth conditions from permissive to non-permissive?

Response 8: We apology for confusions. We performed two successive rounds of 1-liter geraniol fermentation. Both the multiplex safeguard strain and the wild-type strain were modified to be able to produce geraniol for fermentation. The modified multiplex safeguard strain was used in the first round of fermentation for 72 h. After draining the fermentation broth and washing cells with 3-liter sterilized water, the second round of fermentation was carried out by the modified wild-type strain grown in the non-permissive medium for the safeguard strain. Therefore, the safeguard strain was significantly diluted after cell washing, and there were only ~8000 CFU of safeguard strain per ml retained at the initial point of the second round of fermentation. In order to demonstrate the application of our multiplex biocontainment strategy to protect proprietary strains and prevent contamination, we monitored the dynamic change of the residual safeguard strain during the second round of geraniol fermentation using a wild-type strain derivative after switching growth conditions (from permissive to non-

permissive culture medium for safeguard strain). Actually, the second round of fermentation could use any strain of interest, and we used a wild-type strain just as a demo. We modified the relevant paragraph (in line 362 to 380) and Fig. 6 to better describe the experiment design and how we conducted the real fermentation experiment. In addition, we repeated the fermentation experiment two more times to make our results more solid, and the relevant data are also updated.

- The authors have based their system on the LeuOmeRS/tRNACUA system derived from the EcLeuRS for incorporation of O-methyl-L-tyrosine (OMeY). First, since the unAA is OMeY, is the name of the aminoacyl-tRNA synthetase OMeYRS instead of OmeRS? Second, the fidelity of the orthogonal translation systems are always a concern for synthetic auxotrophy applications, especially when the essential proteins are not computationally designed to accommodate only an unAA. This is a highly pertinent question as it seems the authors have consistently used a Leucine dropout media for their experiments. Their use of the Leu marker for cloning justifies the use of a Leucine dropout media for strain construction but not for escape assays, particularly as this manuscript attempts to describe a real fermentation scenario. In real fermentation or the environment, leucine or tyrosine (structurally similar to OMeY) may be present in higher abundance. The authors should conduct experiments with at least the strains that are not reported to escape (or the strains that have low detectable escape rates) to show to readers whether the system breaks in the presence of 1-5 mM Leucine or 1-5 mM tyrosine. In my opinion, if marginally higher escape rates are detected then this is not a major barrier to publication in this journal given how well the system works in the dropout condition, but it is vital to share how context-dependent the observed behavior is in regard to amino acid composition.

Response 9: The name of LeuOmeRS is referring to the engineered leucyl tRNA-synthetase mutant from *Escherichia coli* to genetically encoded OMeY and this name has been commonly used in the field (PMID: 30139255, 36346914 and 36192484). Thus, we used the same name to be consistent with previous works.

To answer the reviewer's concern about the fidelity of LeuOmeRS/tRNACUA system in this work, we swapped the Leu maker to His marker on the plasmid encoding LeuOmeRS/tRNACUA and performed additional escape assay experiments using two representative strains (CDC27_{TAG5} and CDC27_{TAG5}-RPS3 that have low detectable and undetectable escape rates, respectively) grown in 5 mM leucine and 2.6 mM tyrosine (the maximal solubility in water at physiological pH and room temperature). We found CDC27_{TAG5} exhibited close escape frequency under 2.6 mM Tyr and 5 mM Leu conditions (5.1 and 5.5×10^{-7} escapers/CFU respectively at day 4, Fig. S11) compared with the normal condition (8.7×10^{-7} escapers/CFU at day 4, Fig. 1c), and no escaper was observed after plating $\sim 0.8 \times 10^9$ CFUs of multiplex safeguard strain CDC27_{TAG5}-RPS3 on non-permissive plates supplemented with abundant tyrosine and leucine (Fig. S11). Thus, our data suggests that our biocontainment system would still be robust in a real fermentation scenario or the environments, where leucine or tyrosine may be

present in higher abundance. A new paragraph was added to show relevant results in line 341-354.

- When coupling synthetic auxotrophy to the galactose kill switch, the authors should provide more explanation of the intended uses or limitations of the kill switch. It does not seem like a particularly useful technology because it leads to cell death if glucose is around which is not a great trait to have, either for a real fermentation (given how glucose is usually the desired carbon source) or for accidental release.

Response 10: Thank you for this suggestion, a discussion regarding this limitation is now added in line 443-446.

Minor Comments

- In the introduction, the authors should briefly clarify that the complete Sc2.0 strain has not yet been created but that biocontainment is an important goal for that strain.

Response 11: Thank you for this comment. The introduction is modified to clarify this point (line 83-85).

- line 47: when first describing the GMO escapee rate, the authors should specify that this is the maximum acceptable rate of escape from intrinsic biological containment, and that the units presented are escapees per colony forming unit.

Response 12: The introduction is modified according to the reviewer's suggestion (line 50-52).

- Lines 95-98: When the authors revise their description of the real fermentation experiment, they should ensure that this sentence contains more clarifying language about what they tested.

Response 13: The relevant description is now clarified and expanded (line 99-104).

- Lines 159-161: The authors claim that there is no growth without OMeY but then escape rates are provided (10^{-5} - 10^{-8}). The same issue arises later for Fig 2b. Could the authors clarify what they mean by no growth in these cases? Also, the non-diluted samples in the serial dilution images appear faintly. Is this a film reflective of outgrowth that never becomes more dense, or do the strains escape when plated at high density? It would be helpful if the authors specify at what time the images of the serial dilutions are taken and how long the observation period is for these cases, because it may be the case that at high density of plating the escape rates are different.

Response 14: Please note that we used the plate-serial dilution spotting images (Fig. 1b and Fig. 2a) to simply show the effectiveness of the unAA-dependent biocontainment strategy but not for measurement of escape rates. Instead, we

performed separate escape assays for each strain using much larger volume of cells to measure their escape rates as mentioned in response 3 and method section. For the non-diluted samples (the first column) in the serial dilution images, around the $6-9 \times 10^4$ cells (based on the OD_{600} after rough calculation) were dropped onto the non-permissive medium plates. Considering the measured escape rates of these strains (10^{-5} - 10^{-8}), it makes sense that we did not observe escapers in the spotting assay (Fig 1b). We rephrase the relevant description “strains grew well on the medium plate containing 1 mM OMeY and showed prohibited growth in the absence of OMeY compared with the wild-type control strains” in the updated manuscript to better clarify this point.

The reason why the non-diluted samples (first column) appear faintly is probably due to the residual OMeY (please note the synthetic auxotrophs were grown in the permissive medium before plating), which allows the cell growth to a certain extent. Actually, they never became denser and we did not detect any cell growth for the diluted columns, while the wild-type strain grew well (Fig. 1b and Fig. 2a). Please note this phenomenon (weak growth of the non-diluted samples on the non-permissive plates) was also observed in other studies that develop small molecule-dependent biocontainment system using budding yeast (PMID: 28174266, 25624482). To follow the reviewer’s suggestion, we specify at what time the images of the serial dilutions are taken in the updated figures.

- Lines 174: Could the authors provide more explanation of how they chose these 4 genes?

Response 15: We chose these four genes because their expression levels are higher than Cdc4/27 with varying degree. In addition, the ts mutants for these four genes are available, which would be very helpful if we had troubles to construct the corresponding OMeY-dependent strains.

- Lines 166, 177: When values of escape or escape rates are provided, they should be followed by these units (escapes/CFU or escapers/CFU). Escapes would be preferable language based on literature precedent.

Response 16: We have corrected in the revised manuscript.

- Line 117: It would be helpful in this paragraph to inform the reader what the essential functions of CDC4 and CDC27 are, and why they could not be complemented by cross-feeding of metabolites.

Response 17: The function of CDC4 and CDC27 are provided in the updated manuscript (line 124-127).

- Line 173: "series"

Response 18: We have corrected in the revised manuscript.

- Line 181: “The *ts* strain-based spotting assay suggested OMeY incorporation efficiency by LeuOmeRS/tRNA_{CUA} might not be enough to produce sufficient amount of Nmd3 protein for cell survival (Fig. S4).” This seems like the most likely reason why cell survival might not occur, but the spotting assay does not directly indicate anything about the amount of Nmd3 protein being produced. If the authors wish to make this statement they would need to use a different kind of assay technique. Instead, the authors could say that “The spotting assay indicated that cell survival in permissive media at high temperatures was only faintly observed in conditions of limited or no dilution (Fig. S4), which is likely because insufficient Nmd3 protein was produced.”

Response 19: As suggested by another reviewer, we performed additional experiments to screen permissive sites of Nmd3 and found residues 4, 6, 10, 11 and 13 are tolerant of substitution. We also observed all these auxotrophs exhibited obvious growth defect in varying degrees (Fig. S4a). To further understand this, we constructed strains that encode Nmd3–GFP fusion proteins (in both wild-type strain and NMD3_{TAG13}, the most robust auxotroph) on the chromosome and measured fluorescence signals of cells to quickly evaluate the expression level of Nmd3 protein. We found the fluorescence intensity of the strain NMD–GFP is at around 7-fold higher than that of NMD_{TAG13}–GFP (Fig. S4b), which suggested the impaired growth of the *NMD3*-based auxotroph is caused by insufficient production of Nmd3 protein by LeuOmeRS/tRNA_{CUA} mediated amber suppression. We have revised the relevant descriptions in line 189 to 197 and relevant Figures.

- When escape was observed from the temperature sensitive mutant screen, could the authors shed more light on why that may be occurring?

Response 20: We agree with the reviewer that it is an interesting observation, but understanding of the escape mechanism for *ts* mutants is not the main focus of this paper as we only use *ts* mutants for screening and quick validation. We would further look at this by sequencing the escapers in following studies.

- Line 186: The claim about transcriptional rates seems difficult to justify when a gene corresponding to only one high RKPM value (300) did not function well but the gene with the next highest RKPM value (194) did function well. Either the claim should be softened or genes with more RKPM values in the 200-350 range should be tested to see if this is a real trend rather than one unique case.

Response 21: We now obtained OMeY auxotrophs based on *NMD3* (RPKM at 305) and showed synthetic auxotrophs were successfully constructed based on all 7 selected genes with very distinct expression levels (Fig. S1). We have also rephrased our claim accordingly in the revised manuscript from line 197 to 200.

- Line 190: More description around this section about how the escape assay was conducted may help readers newer to this concept.

Response 22: We provide detailed description about the escape assay in the method section and point it out in the updated manuscript.

- Line 212: Please clarify that this improved fitness is in non-permissive conditions.

Response 23: Thank you for this suggestion. We have added accordingly in the updated manuscript.

- Line 243: In the text it would be helpful if the authors could list the escape rates obtained with and without the use of the CRISPR-Cas9 system, or at least representative or best values.

Response 24: Thank you for this suggestion. The best values are added in the updated manuscript in line 261-264.

- Line 249: it wasn't clear what these leftover scar interrupts were. are these mutations that did not disrupt the PAM sequence but that altered the adjacent region to create mismatches to the guide RNA?

Response 25: Yes, it is just as what the reviewer described. We showed these mutations in detail in Fig. S6.

- Line 259: How did the authors settle on these genes?

Response 26: We chose *RPC11* gene because transcriptional regulated safeguard strain based on this gene was successfully constructed in a previous study (PMID: 25624482). We also tested two additional genes, namely *SKP1* and *RPS3* (RPKM values at 371 and 1329 respectively), that exhibit much higher transcription levels than *RPC11* (RPKM value at 34); and their transcriptional level are in top 10% of all essential genes (Fig. S1). As described in the manuscript, we chose these two genes because we hypothesized that leaky expression of highly expressed genes should not be enough to support the cell growth under the non-permissive condition. In the updated manuscript, we included the RPKM value for these selected genes to help the readers better know the difference of their transcription levels.

- Line 280: The conclusion for the doubling time data here is inconsistent with the data observed

Response 27: We have rephrased this sentence to describe the doubling time data more accurately.

- Lines 287-288- What was the motivation here if, in the previous paragraph, the authors say there was not fitness impairment?

Response 28: We have rephrased the relevant sentence in the updated manuscript (line 308-309).

- Line 300: "revealed"

Response 29: We have corrected it accordingly.

- Line 330: Avoid the word "significantly" unless stating statistical significance

Response 30: We have removed it accordingly.

- Line 333: Could the authors state what G418 is? Seems like an antibiotic but it is unclear.

Response 31: We have added the description of antibiotic before G418 in the updated text.

- Line 371: Revise to make clearer that this statement is about the past, before this study: "The escape mechanisms that disable the unAA-dependent biocontainment system in *S. cerevisiae* were previously unexplored."

Response 32: We have revised this statement as suggested.

- Lines 402-404: The authors' perspective on the difficulty of identifying how to grow a synthetic auxotroph would be far better supported if they could show that their synthetic auxotrophs are not capable of growing on other unAAs. It is widely known that many orthogonal translation systems exhibit polyspecificity. Earlier, this reviewer pointed out that the growth on media containing Leu or supplemented Tyr should be tested. For claims such as these, the authors should consider testing other unAAs, or they should temper their remarks.

Response 33: To follow the reviewer's suggestion, we now temper our remarks in the corresponding discussion (line 460-462). As mentioned in response 9, we measured escape frequencies under conditions containing high concentration of Leu and Tyr, and close escape rates were detected under these conditions.

- Line 443: typo for "safeguard"

Response 34: We have corrected it accordingly.

- Line 455: It is unclear how the analysis of conserved residues was performed and the subsequent data representation in the extended data figures. Was the conservation analysis performed across all essential proteins in yeast? Were those genes compared against specific orthologs from the BLAST search? Was there a cutoff for the similarity of the ortholog?

Response 35: Yes, the conservation analysis was performed across all essential proteins in yeast by comparing against specific orthologs from the BLAST search. The cutoff for the similarity of the ortholog was set at 0.01. We revised the corresponding method section according to the reviewer's comments.

- Line 476: typo for "varied"

Response 36: We have corrected it accordingly.

- Line 518: typo for "medium"

Response 37: We have corrected it accordingly.

- Line 520: Please clarify the suspected dilution for plating on permissive media? What is the precedence for approximating CFU counts with this method?

Response 38: Please refer to response 3 and revised "Escape assays" method section for details.

- Line 584: the number for k seems to be incorrect, or at least the comma is placed at the wrong location

Response 39: We have corrected it accordingly.

- I may have missed this, but when two TAG codons are used together in the same gene, or when the transcriptional and translational control systems are used together, what are the mechanisms of escape when escape is observed?

Response 40: Please note that we only constructed a synthetic auxotroph (CDC27_{TAG520}CDC4_{TAG9}) that contain two TAG codons in different genes. Since we did not observe any escaper derived from the two-layered safeguard strains under our detection limit ($1.1-1.2 \times 10^{-9}$), we could not go further to analyze the potential escape mechanisms.

- Figure 1: the doubling time plot in panel C does not have the bar graphs labeled on the x-axis. one would think that maybe the labels on the bottom plot in panel C could apply to the top plot, but there are 10 bars in the top plot and only 9 strains shown on the bottom plot. my guess is that the first 9 bars correspond to these strains in the

same order, and that the last bar is the WT strain, but I should not have to guess. please provide labels in the top plot.

Response 41: Thank you for pointing it out. Yes, the first 9 bars correspond to these strains in the same order, and the last bar is the WT strain. We removed the last bar (corresponding to WT strain) on the top plot to avoid confusion, and the top and bottom plot could share the same x-axis.

- (this is just a comment, no corrective action is required) Figure 2: while the serial dilutions shown in 1b, 2a, and throughout the SI do not provide much quantitative information, I appreciate their inclusion as it shows the reliability of the serial dilutions. Presumably, larger volumes of the same serial dilution samples were used for the measurement of escape frequency.

Response 42: As answered in response 14, the plate-serial dilution spotting images (Fig. 1b and Fig. 2a) was used to simply show the effectiveness of the unAA-dependent biocontainment strategy but not for measurement of escape rates. Instead, we performed separate escape assays for each strain to measure their escape rates using much larger volumes of the cells. Please see the source data file to find detailed and quantitative information that were used to calculate escape rate as mentioned in response 3.

- Figure 2: bottom plot in 2b should state that this was escape frequency seen at 2 days, independent of whether figure 1 states this or the figure caption states this

Response 43: Actually, the escape frequencies corresponding to bottom plot in 2b were taken on the day 8 as mentioned in the figure legend. We now state this in the updated figure 2 to follow the reviewer's suggestion.

- Figure 3: In figure 3a, it is unclear why the nearly identical image is shown twice - the "no cleavage", "normal strains", "gene encoding tRNATyr". The only difference is the inclusion or not of Cas9 and guide, but it feels like the information is redundant. The concept could be better illustrated by showing the wild-type strain and sequence first with this gRNA and SpCas9-NG unbound to the wild-type sequence, and to then have an arrow showing that if an escaper arises within the population then the image that is currently the first image on the left would apply. Note that this image is currently somewhat redundant with the third image in this series, so if the authors were to follow these suggestions then they would simplify the image from 4 to 2.

Response 44: Thank you for the comments, we now modified Fig. 3a according to the reviewers' suggestions.

- Figure 4: Figure 4b confuses me. According to Fig. 4a and the manuscript text, the 2% glucose condition should be non-permissive and the 2% galactose condition

permissive, correct? If that is the case then Figure 4b has this labeled backwards. Also, the depiction of serial dilution for the transcriptional kill switch raises the question of what the escape rates were since this does not seem to work as well as synthetic auxotrophy. that is consistent with past literature but would be good to reinforce. were the escape rate measured?

Response 45: Sorry for the confusion. The label for galactose and glucose conditions in Fig. 4b is mistakenly swapped. We have corrected this error in updated Fig. 4. We indeed measured the escape rates of all three strains with transcriptional kill switch as shown in Fig. S7.

- Figure 6: As mentioned in the Major Comments section, I strongly suggest revising this figure to provide more conceptual depiction of the experiment. I also think the inclusion of the probabilities above the last two bars is distracting and unnecessary – these values can simply be referenced in the text.

Response 46: We have revised the Fig. 6A to help the reader better understand how the experiment was conducted. We removed the probabilities above the last two bars in Fig. 6 according to the reviewer's suggestion.

- When the authors comment on a real fermentation scenario, they should provide readers with a better sense of the total CFU expected of yeast cells at different scales of fermentation, and they should also provide some justification for why they have chosen to culture only $\sim 10^9$ cells since prior studies have reported lower assay detection limits – in many cases 2 or 3 orders of magnitude lower. Practically speaking, plating 10^9 cells is more convenient, especially if the goal is to be below the NIH biocontainment standard. However, for real fermentation one would want to know if the strains that do not exhibit observable escape in this study are actually suitable or if they simply escape at 10^{-10} escapees per CFU.

Response 47:

As mentioned in the third paragraph of response 3, we believe our assay limit is reasonable and reaches the high standard set according to literatures about yeast-based biocontainment system. Actually, the main purpose for us to perform a real fermentation scenario is to demonstrate the application of our multiplex biocontainment strategy for: (i) restricting the chance of stealing strains by recovering from a small amount of residual fermentation broth and (ii) preventing contamination in the fermentation process due to residual cells survived from incomplete sterilization (as mentioned in the revised manuscript in line 357-373). Different from the escape assays, the fed-batch fermentation experiments allowed us to monitor the dynamic change of the residual safeguard strain during a cycle of 1-liter fermentation using a wild-type derived strain after switching growth conditions. By doing this, we could determine the residual CFUs and proportion of safeguard strain, which allowed us to calculate the probability of successfully sub-culturing the safeguard strain from the broth. Thus, we

believe this data obtained from our real fermentation assay would be valuable and show the potential of biocontainment system to prevent stealing of industrial strains, which is now receiving widespread attentions in the industrial field of synthetic biology.

Reviewer #2 (Remarks to the Author):

Chang et al. presented a novel two-layered approach for the construction of a biocontainment system. The first layer involves the implementation of an unnatural amino acid dependent gene expression system by incorporating an amber codon at the permissive site of an essential gene. The second layer utilizes a transcriptional control mechanism through the installation of a galactose-inducible promoter, enabling the expression of another essential gene.

In addition to leveraging a well-established amber suppression strategy previously employed in bacterial cells, the authors developed a galactose-inducible gene expression system to enhance the stringency of the biocontainment system in eukaryotic cells.

While the manuscript is generally well-written, I would suggest revising the following aspects:

Response 1: We thank the reviewer for in-depth and comprehensive interpretation of our work and for recognizing the values of our study.

1. [Page 5, line 122] References 15 and 16, which are also related to the "design" approach, should be included alongside the other references.

Response 2: We have added these references accordingly.

2. [Pages 5 and 6, lines 121–133] [Pages 13 and 14, lines 364–367] The strategy of identifying permissive sites has been extensively studied in previous research (ref. 12, Rovner et al., Nature 2015). The authors should compare the two algorithms and explain the advantages of the current approach over the previous one.

Response 3: We actually did not develop a specific algorithm to identify permissive residues, instead, we simply look for potential permissive residues that show low conservation among protein homologs, with particular interest in the N-terminus, by BLAST search (described in the method section). Indeed, our idea is inspired by ref 12, but please note that ref 12 also did not develop their own algorithm but used SIFT algorithm (ref 37). In this study, we also used SIFT algorithm to find additional permissive residues in the rest region of Cdc4 and Cdc27 proteins. We revised relevant descriptions in the updated manuscript (in line 130-140) to better acknowledge the contribution of ref 12 and explain our approach to identify permissive residues. In addition, we further discussed the advantage of our library-based screening method in the modified discussion section (line 415-417).

3. [Page 7, line 173] "serious" → "series"

Response 4: We have corrected it accordingly.

4. [Page 7, lines 180–183] It is necessary to clarify whether the mutation site of NMD3 is permissive. Additionally, what is the minimal expression level of NMD3? Overall, more experiments are needed to confirm whether the inability to obtain an OMeY auxotroph by targeting NMD3 is caused by an insufficient amount of Nmd3 protein. For example, determining the minimal expression level of Nmd3 for cell survival, testing the suppression efficiency of the LeuOMeRS/tRNA system, and screening for permissive sites of Nmd3.

Response 5: To follow the reviewer's suggestion, we performed additional experiments to screen permissive sites of Nmd3 and found residues 4, 6, 10, 11 and 13 are tolerant of substitution. We also found all these auxotrophs exhibited obvious growth defect in varying degrees (Fig. S4a). To further understand this, we constructed strains that encode Nmd3–GFP fusion proteins (in both wild-type strain and NMD3_{TAG13}, the most robust auxotroph) on the chromosome and measured fluorescence signals of cells to quickly evaluate the expression level of Nmd3 protein. We found the fluorescence intensity of the strain NMD–GFP is around 7-fold higher than that of NMD_{TAG13}–GFP (Fig. S4b), which suggested the impaired growth of the NMD3-based auxotroph is caused by insufficient production of Nmd3 protein by LeuOmeRS/tRNA_{CUA} mediated amber suppression. We revised the relevant descriptions (line 189-197) and included the new data in updated Fig. 2 and Fig. S4.

5. [Page 7, lines 185–186] The statement "has good versatility for many different essential proteins in *S. cerevisiae*" is exaggerated, considering that it only includes the group of low-expression genes.

Response 6: This claim is now rephrased in the updated text based on our new data in line 197-200.

6. [Page 8, line 201] [Page 14, line 374] "Si, L. et al. (2016) Generation of influenza A viruses as live but replication-incompetent virus vaccines. *Science*, 354, 1170–1173." should be included in the references.

Response 7: We have added this reference accordingly.

7. [Page 8, line 206] The authors should discuss other possibilities of misincorporation of natural amino acids by LeuOmeRS/tRNA in the absence of OMeY.

Response 8: We have added this statement accordingly.

8. [Page 9, lines 221–222] While the CRISPR-Cas9 strategy is a good idea, it had little

impact on the biocontainment system, possibly due to its limited targeting on tRNA-Tyr, not other escape mechanisms. This limitation should be discussed, along with future directions for this "immunity" based approach.

Response 9: Limitations along with future directions of this approach are now added in the discussion section in line 429-434.

9. [Page 10, lines 267, and Figure 4b] The images for galactose and glucose conditions appear to be swapped.

Response 10: Thank you for pointing this out. We have corrected it accordingly.

10. [Page 12, line 307] "reside" → "residues"; "residue 7,8,11,14" → "residues 7, 8, 11, and 14"

Response 11: We have corrected them accordingly.

11. [Page 12, lines 335] While it is true that more than 200 unnatural amino acids can be genetically encoded, it should be confirmed how many of these can be encoded in yeast, as most host organisms are bacteria, not yeast.

Response 12: The reviewer's raised a good point. We now have changed this number to ~60 based on a review paper published in 2016 (PMID: 26868890) and all relevant literatures coming after this paper.

12. The transcriptional-based switch does not seem to be a perfect biocontainment system because unlike unnatural amino acids, galactose is naturally available. The authors should include a discussion regarding this limitation.

Response 13: Thank you for this suggestion, a discussion regarding this limitation is now added in line 443-446.

Reviewers' Comments:

Reviewer #1:

Remarks to the Author:

The authors have satisfactorily addressed each of the comments raised during the review process.

Reviewer #2:

Remarks to the Author:

The authors addressed the issues raised by the reviewers, providing clarification and including additional experiments on amber suppression efficiency, screening for new permissive sites in Nmd3, and fidelity tests using high concentrations of tyrosine and leucine. Moreover, the authors made specific corrections and integrated the recommended references, which has enhanced the manuscript's overall quality. Their discussion about the limitations, particularly concerning the transcriptional-based switch, offers a broader perspective on the paper. Given these revisions, I am inclined to believe the manuscript is now better poised for publication.

Below are some minor questions and potential typos to consider:

1. [Line 139] Should "computationally" be changed to "computational"?
2. [Line 190] Did you intend to write "new" instead of "few"?
3. The terms "escaper" and "escapee" are both used throughout the text. For consistency, I recommend choosing one, with "escapee" being my suggestion.

We would like to thank the reviewers for their time and suggestions that helped us improve our work. Below, we provide a point-by-point response (in blue color) to reviewers' comments.

Reviewer #1 (Remarks to the Author):

The authors have satisfactorily addressed each of the comments raised during the review process.

Response: We thank the reviewer for recognizing the value of our study.

Reviewer #2 (Remarks to the Author):

The authors addressed the issues raised by the reviewers, providing clarification and including additional experiments on amber suppression efficiency, screening for new permissive sites in Nmd3, and fidelity tests using high concentrations of tyrosine and leucine. Moreover, the authors made specific corrections and integrated the recommended references, which has enhanced the manuscript's overall quality. Their discussion about the limitations, particularly concerning the transcriptional-based switch, offers a broader perspective on the paper. Given these revisions, I am inclined to believe the manuscript is now better poised for publication.

Response: We thank the reviewer for recognizing the value of our study.

Below are some minor questions and potential typos to consider:

1. [Line 139] Should "computationally" be changed to "computational"?

Response: Thank you, we have corrected it accordingly.

2. [Line 190] Did you intend to write "new" instead of "few"?

Response: We removed "few" to avoid confusions.

3. The terms "escaper" and "escapee" are both used throughout the text. For consistency, I recommend choosing one, with "escapee" being my suggestion.

Response: Thank you for this suggestion, we now only use "escapee" throughout the text in the updated manuscript.